# Courtroom Analogy: New Perspective on Uncertainty-Aware Classification

**Taeseong Yoon** [1]   **Heeyoung Kim** [1]

## Abstract

Single-pass uncertainty quantification (UQ) methods for classification represent uncertainty by predicting a tractable distribution over the class probability vector. While existing approaches primarily focus on enhancing the expressiveness of this distribution, they often provide limited insight into how predictive uncertainty is structured and aggregated, resulting in weak interpretability. We introduce the *courtroom analogy*, which conceptualizes uncertainty-aware classification as a structured debate among class-specific advocates. Each advocate forms a probabilistic opinion, and a final verdict is reached by aggregating these opinions using input-dependent plausibility weights. In this framework, each advocate's opinion is modeled as a Dirichlet distribution whose concentration parameter is decomposed into shared evidence and class-specific advocacy. This yields a structured mixture of Dirichlet distributions with semantically interpretable parameters. To instantiate this formulation, we propose *Mixture of Dirichlet EXperts* (MoDEX), a single-pass neural architecture that predicts the courtroom parameters, enabling efficient and expressive UQ while explicitly modeling uncertainty aggregation. We demonstrate that MoDEX enjoys strong theoretical properties and achieves state-of-the-art UQ performance across diverse benchmarks, yielding interpretable uncertainty estimates with meaningful semantics.

## 1. Introduction

Recent advances in machine learning models have delivered remarkable predictive accuracy across a wide range of domains. However, their deployment in high-risk and safety-critical applications remains limited due to reliability concerns—most notably, the tendency of modern neural networks to produce overconfident predictions that fail to reflect predictive uncertainty (Guo et al., 2017). This limitation underscores the need for principled uncertainty quantification (UQ) methods that can reliably quantify predictive uncertainty alongside their predictions. For practical deployment, such UQ methods must be computationally efficient and provide uncertainty estimates with clear probabilistic semantics tied directly to the predictive distribution. These considerations have motivated the development of robust *single-pass* UQ methods (Sensoy et al., 2018), which estimate predictive uncertainty in a single forward pass without relying on sampling-based or computationally intensive components.

A promising direction toward single-pass UQ is to directly predict a *second-order predictive distribution*, which in the classification setting corresponds to a distribution over class probability vectors. This formulation explicitly captures uncertainty at the level of the predictive distribution, as exemplified by evidential deep learning (EDL) (Sensoy et al., 2018). A key design consideration is the output distribution family $\mathcal{Q}$ defined over the appropriate output space (e.g., the probability simplex in classification). An effective choice of $\mathcal{Q}$ must satisfy: (i) *finite and tractable parametrization*, such that each $q \in \mathcal{Q}$ admits a finite-dimensional representation with closed-form predictive summaries for efficient single-pass inference; and (ii) *expressiveness*, such that $\mathcal{Q}$ has sufficient capacity to represent diverse uncertainty patterns encountered in practice.

Under these criteria, much of the second-order UQ literature adopts Dirichlet-type families for $\mathcal{Q}$ due to their finite and tractable parametrization, with EDL serving as a canonical example (Sensoy et al., 2018). Subsequent variants retain the Dirichlet choice for $\mathcal{Q}$ while introducing modifications to improve the expressiveness of uncertainty representations (Deng et al., 2023; Yoon & Kim, 2024; Chen et al., 2024; 2025). More recent work, such as $\mathcal{F}$-EDL (Yoon & Kim, 2026), extends the Dirichlet family itself by introducing a more expressive yet still tractable distribution for $\mathcal{Q}$. Collectively, these efforts reflect a common trend in second-order UQ: enhancing the expressive capacity of $\mathcal{Q}$ while preserving tractable inference. However, this expressivity-driven line of work provides limited insight into how predictive un-

[1]Department of Industrial and Systems Engineering, Korea Advanced Institute of Science and Technology, Daejeon, South Korea. Correspondence to: Heeyoung Kim <heeyoungkim@kaist.ac.kr>.

*Proceedings of the 43rd International Conference on Machine Learning*, Seoul, South Korea. PMLR 306, 2026. Copyright 2026 by the author(s).

certainty is structured and aggregated within $\mathcal{Q}$, resulting in uncertainty representations with largely implicit and weakly interpretable semantics.

In this work, we introduce a new perspective on uncertainty-aware classification by explicitly encoding how predictive uncertainty is formed and aggregated through the design of a structured output family $\mathcal{Q}$. To this end, we propose a novel *courtroom analogy* that provides a principled and semantically grounded framework for understanding and modeling the process of uncertainty-aware classification.

Specifically, under this framework, an input $\mathbf{x}$ corresponds to a *case* presented to the court, and classification is framed as a structured debate among $K$ *advocates* in the courtroom, each representing one of the $K$ possible labels. Although all advocates examine the same case evidence, they may arrive at distinct probabilistic beliefs over the class probabilities. The final *verdict* is modeled as a mixture of advocates' opinions, whose mean and dispersion correspond to the point prediction and predictive uncertainty, respectively. Crucially, this formulation explicitly represents multiple sources of predictive uncertainty, enabling it to accommodate diverse input regimes in which uncertainty may stem from insufficient evidence, conflicting interpretations of the same evidence, or both.

We instantiate this courtroom-based classification framework through two principled design choices. First, we represent each advocate's opinion using a Dirichlet distribution, consistent with subjective logic (Jøsang, 2016) and evidential reasoning frameworks. Second, we posit that each advocate's opinion admits a structured decomposition into (i) a shared base belief that reflects the objective case facts observed by all advocates, and (ii) a class-specific advocacy strength that modulates the concentration toward the $k$-th verdict, capturing how strongly advocate $k$ supports its class beyond the shared evidence.

These advocates' opinions are then aggregated via input-dependent plausibility weights; mathematically, this aggregation induces a structured mixture of Dirichlet distributions. From a design perspective, the courtroom analogy introduces a new paradigm for constructing $\mathcal{Q}$: uncertainty is not merely represented, but structurally induced through interpretable, class-wise opinions and their aggregation. This perspective departs from prior expressivity-driven extension of Dirichlet models and instead elevates structural inductive bias as the primary mechanism for uncertainty modeling.

As a concrete realization of this design principle, we introduce the *Mixture of Dirichlet EXperts* (MoDEX), a neural architecture that provides a direct instantiation of the courtroom analogy for uncertainty-aware classification. Given an input, MoDEX predicts class-specific Dirichlet opinions whose concentration parameters are each decomposed into

shared evidence and class-dependent advocacy, together with input-dependent plausibility weights that govern their aggregation. All components are learned end-to-end within a unified neural framework. We further develop a tailored training objective and principled uncertainty measures that enable reliable prediction and UQ within this framework.

This design offers four key benefits: (i) *efficiency* via closed-form moments of the Dirichlet distribution; (ii) *parameter efficiency* from the structured courtroom modeling; (iii) *expressiveness* from the flexible distributional modeling afforded by a structured mixture of Dirichlets; and (iv) *interpretability* through the courtroom semantics of the parameters. Moreover, MoDEX's probabilistic formulation coincides with the extended Flexible Dirichlet (EFD) family (Ongaro et al., 2020), providing a principled characterization of the shared vs. class-specific concentration structure.

MoDEX offers several theoretical advances that establish it as a principled UQ framework. First, we prove that MoDEX reduces to prior methods, including $\mathcal{F}$-EDL (Yoon & Kim, 2026) and standard EDL (Sensoy et al., 2018), under specific parameter settings, thereby positioning MoDEX as a principled generalization of existing models. Second, we prove that MoDEX admits two equivalent interpretations: (i) a weighted ensemble of $K$ EDL experts with expert-specific evidence, and (ii) a mixture of an EDL expert with shared base evidence and a softmax predictor. These perspectives suggest that MoDEX can dynamically adapt to different input regimes by combining complementary uncertainty reasoning mechanisms. Third, we prove that MoDEX's epistemic uncertainty decomposes into intra-expert and inter-expert components, capturing multiple sources of epistemic uncertainty and demonstrating its rich uncertainty modeling capacity.

MoDEX consistently achieves state-of-the-art performance on various UQ-related downstream tasks across diverse ID regimes, while enabling a semantically interpretable uncertainty-aware classification process.

## 2. Related Work

Second-order UQ methods represent predictive uncertainty by learning a distribution over the latent class probability vector. We categorize them into *explicit* methods, which directly parameterize a distribution over class probabilities, and *implicit* methods, which induce second-order uncertainty via auxiliary randomness or structural constraints.

**Explicit Second-Order UQ Methods.** A representative explicit approach is EDL, which predicts a Dirichlet distribution over the class probability vector in classification tasks (Gao et al., 2025). EDL was first introduced by Sensoy et al. (2018), with connections to Dempster-Shafer Theory (Dempster, 1968) and subjective logic (SL) (Jøsang, 2016).

Under the lens of SL, Dirichlet concentration parameters can be interpreted as evidence accumulated for categorical outcomes, providing an intuitive basis for uncertainty-aware prediction. Since then, several works have extended EDL in various directions. $\mathcal{I}$-EDL (Deng et al., 2023) incorporates Fisher information to capture data-dependent uncertainty. R-EDL and Re-EDL (Chen et al., 2024; 2025) relax nonessential assumptions of the original formulation, while $\mathcal{F}$-EDL (Yoon & Kim, 2026) replaces the Dirichlet distribution with a more flexible alternative to improve expressiveness. Additional efforts enhance epistemic UQ via auxiliary OOD data (Sensoy et al., 2020), using density estimation (Charpentier et al., 2020; 2021; Yoon & Kim, 2024), or integrating knowledge distillation (Malinin et al., 2019; Wang & Ji, 2024; Shen et al., 2024). Despite these advances, most methods focus on enriching second-order distributions or training signals, while leaving the structural mechanism by which predictive uncertainty is formed and aggregated largely implicit. In parallel, recent work has questioned the meaningfulness of epistemic uncertainty learned solely from one-hot supervision (Bengs et al., 2022; 2023; Juergens et al., 2024; Shen et al., 2024).

**Implicit Second-Order UQ Methods.** Beyond explicitly parameterizing second-order distributions, several UQ approaches can be interpreted as implicitly inducing second-order uncertainty. Bayesian UQ methods (Blundell et al., 2015; Gal & Ghahramani, 2016; Kristiadi et al., 2020; Harrison et al., 2024) place a distribution over neural network weights, which in turn induces a distribution over the predicted class probabilities. Similarly, Deep Ensembles (Lakshminarayanan et al., 2017) induce predictive uncertainty by aggregating predictions from multiple independently trained models. However, these approaches typically rely on multiple forward passes at inference, which limits practical efficiency. Deterministic UQ methods (Van Amersfoort et al., 2020; Liu et al., 2023; Mukhoti et al., 2023; Bui & Liu, 2024) instead impose geometric or distance-aware constraints in feature space, yielding proximity-based uncertainty scores rather than an explicit distribution over class probabilities, limiting second-order interpretability.

# 3. Courtroom Analogy for Uncertainty-Aware Classification

We introduce the *courtroom analogy*, a principled and interpretable framework for uncertainty-aware classification. Under this perspective, classification is framed as a structured courtroom debate among $K$ advocates—one for each class—who evaluate a *case* $\mathbf{x}$ and present competing arguments, with the final *verdict* given by a class label $y \in \{1, \ldots, K\}$. Although all advocates observe the same case evidence, they may emphasize different aspects of it, leading to distinct probabilistic beliefs about each verdict. In this way,

the courtroom debate naturally gives rise to rich forms of predictive uncertainty, reflecting both the uncertainty inherent in each advocate's belief and the disagreement among advocates evaluating the same case.

In the following, we formalize the generative process underlying the courtroom analogy and introduce interpretable modeling design choices. We then show that the resulting distribution over class-probability vectors can be expressed as a structured mixture of Dirichlet distributions.

## 3.1. From Debate to Distribution: A Generative Perspective

Let $\mathcal{D} = \{(\mathbf{x}_i, y_i)\}_{i=1}^N$ denote a labeled dataset, where each input $\mathbf{x}_i \in \mathcal{X}$ represents a *case* and each label $y_i \in \{1, \ldots, K\}$ corresponds to a possible *verdict*. Uncertainty-aware classification is modeled as a structured debate among $K$ advocates, each representing one of the possible classes. As in standard multiclass classification, an advocate is assigned to each verdict, even when certain verdicts are implausible for a given case.

For each input $\mathbf{x}_i$, we introduce a latent class-probability vector $\boldsymbol{\pi}_i = (\pi_{i1}, \ldots, \pi_{iK}) \in \Delta^{K-1}$, where $\pi_{ik} > 0, \forall k \in [K]$ and $\sum_{k=1}^K \pi_{ik} = 1$. This vector represents the underlying categorical distribution over the $K$ possible verdicts for the $i$-th case. Rather than predicting a single deterministic probability vector, we treat $\boldsymbol{\pi}_i$ as a latent random variable and model uncertainty through a second-order predictive distribution defined over $\Delta^{K-1}$.

Each advocate $k$ forms an *opinion* about $\boldsymbol{\pi}_i$, represented by the probability distribution $p(\boldsymbol{\pi}_i \mid L_i = k, \mathbf{x}_i)$, where the latent variable $L_i \in \{1, \ldots, K\}$ indexes the advocate whose interpretation governs the latent class-probability vector for the $i$-th case. The overall, second-order predictive distribution is then modeled as a mixture of these advocate opinions, with mixture weights $\boldsymbol{\omega}(\mathbf{x}_i) = (\omega_1(\mathbf{x}_i), \ldots, \omega_K(\mathbf{x}_i))$, where $\omega_k(\mathbf{x}_i) > 0, \forall k \in [K]$ and $\sum_{k=1}^K \omega_k(\mathbf{x}_i) = 1$. These weights encode the relative plausibility of each advocate's argument for the given case.

Formally, the generative process of each input $\mathbf{x}_i$ proceeds as follows. First, an advocate index $L_i$ is drawn from a categorical distribution with probabilities $\boldsymbol{\omega}(\mathbf{x}_i)$, i.e., $L_i \sim \text{Cat}(\boldsymbol{\omega}(\mathbf{x}_i))$. Conditioned on $L_i = k$, the latent class-probability vector $\boldsymbol{\pi}_i$ is then generated according to the opinion distribution of advocate $k$. Following standard practice in evidential learning (Sensoy et al., 2018), we model each advocate's opinion using a Dirichlet distribution:

$$p(\boldsymbol{\pi}_i \mid L_i = k, \mathbf{x}_i) = \text{Dir}(\boldsymbol{\pi}_i | \boldsymbol{\alpha}_k(\mathbf{x}_i)), \quad \forall k \in [K],$$

where $\boldsymbol{\alpha}_k(\mathbf{x}_i) \in \mathbb{R}_{>0}^K$ denotes the concentration parameter of the Dirichlet distribution representing the advocate $k$'s opinion over $K$ possible verdicts for input $\mathbf{x}_i$. This choice

provides a tractable and interpretable representation of uncertainty over categorical probability vectors. The observed verdict is then drawn from the latent class-probability vector, i.e., $y_i \sim \mathrm{Cat}(\boldsymbol{\pi}_i)$.

Marginalizing out the latent advocate index $L_i$ yields the second-order predictive distribution over class-probability vectors for input $\mathbf{x}_i$

$$p(\boldsymbol{\pi}_i \mid \mathbf{x}_i) = \sum_{k=1}^{K} \omega_k(\mathbf{x}_i)\mathrm{Dir}(\boldsymbol{\pi}_i \mid \boldsymbol{\alpha}_k(\mathbf{x}_i)), \qquad (1)$$

which takes the form of a mixture of Dirichlet distributions, where each component represents an advocate's opinion and mixture weights encode the input-dependent plausibility.

### 3.2. Structured Decomposition of Advocate Opinions

In its raw form, the courtroom mixture in Eq. (1) is parameter-intensive: naively specifying $K$ distinct Dirichlet concentration vectors requires $\mathcal{O}(K^2)$ parameters. To obtain a scalable and interpretable model, we introduce an additional courtroom-inspired modeling approach that imposes structured constraints on advocate opinions, thereby reducing parameter complexity while preserving expressiveness.

For each input $\mathbf{x}_i$, we decompose the concentration vector of each advocate as follows: $\boldsymbol{\alpha}_k(\mathbf{x}_i) = \boldsymbol{\alpha}(\mathbf{x}_i) + \tau_k(\mathbf{x}_i)\mathbf{e}_k$, for all $k \in [K]$, where $\boldsymbol{\alpha}(\mathbf{x}_i) \in \mathbb{R}_{>0}^K$ represents evidence shared across all advocates, $\mathbf{e}_k$ denotes the $k$-th standard basis vector, and $\tau_k(\mathbf{x}_i) > 0$ modulates the additional concentration mass assigned by the advocate $k$ to its own class.

This decomposition separates two sources of influence on an advocate's opinion. The shared base opinion $\boldsymbol{\alpha}(\mathbf{x}_i)$ captures evidence common to all advocates, such as objective facts of the case. In contrast, the class-specific advocacy term $\tau_k(\mathbf{x}_i)$ captures the additional emphasis that advocate $k$ places on class $k$, reflecting interpretive arguments or selective framing that uniquely favor that advocate given the same input. This structured decomposition is intended as an inductive bias, rather than direct supervision of latent roles, encouraging a separation between shared evidence and class-specific emphasis. By clearly defining the role of each parameter, this structured decomposition governs how advocate opinions can diverge, mirroring the deliberative dynamics of a courtroom.

Applying this decomposition to Eq. (1) yields a structured mixture of Dirichlet distributions of the following form:

$$p(\boldsymbol{\pi}_i \mid \mathbf{x}_i) = \sum_{k=1}^{K} \omega_k(\mathbf{x}_i)\mathrm{Dir}(\boldsymbol{\pi}_i \mid \boldsymbol{\alpha}(\mathbf{x}_i) + \tau_k(\mathbf{x}_i)\mathbf{e}_k). \quad (2)$$

In essence, the courtroom-inspired design principles induce a well-defined predictive distribution structured as a constrained mixture of $K$ Dirichlet components, with an $\mathcal{O}(K)$

parameterization and single-pass inference, while also enabling interpretable semantics.

Under this formulation, uncertainty-aware classification reduces to predicting, for a given input $\mathbf{x}_i$, the parameters that define the courtroom-induced predictive distribution in Eq. (2). We address this mapping using a neural network that preserves the semantic roles encoded by the courtroom analogy while enabling principled UQ.

## 4. MoDEX: Mixture of Dirichlet Experts

To operationalize the courtroom analogy within a learnable framework, we propose the *Mixture of Dirichlet EXperts* (MoDEX), a neural instantiation of the courtroom analogy. Given an input $\mathbf{x}_i$, MoDEX employs a neural network to predict the parameters of the structured predictive distribution over the class-probability vector $\boldsymbol{\pi}_i$ defined in Eq. (2), thereby enabling uncertainty-aware classification in a single forward pass. The MoDEX framework comprises three core components: (i) a neural architecture that parameterizes the structured courtroom mixture (Section 4.1), (ii) a tailored learning objective (Section 4.2), and (iii) principled uncertainty measures (Section 4.3).

### 4.1. Neural Instantiation of the Courtroom Analogy

**Neural Parameterization.** Given an input $\mathbf{x}_i \in \mathcal{X}$, a feature representation $\mathbf{z}_i \in \mathcal{H}$ is first computed using a feature extractor $f_{\boldsymbol{\psi}} : \mathcal{X} \rightarrow \mathcal{H}$, i.e., $\mathbf{z}_i = f_{\boldsymbol{\psi}}(\mathbf{x}_i)$. Conditioned on $\mathbf{z}_i$, MoDEX predicts the parameters of a structured Dirichlet mixture that instantiates the courtroom analogy via three lightweight prediction heads: (i) a concentration head $g_{\boldsymbol{\phi}_1} : \mathcal{H} \rightarrow \mathbb{R}^K$ that produces logits for the shared base evidence, (ii) a gating head $g_{\boldsymbol{\phi}_2} : \mathcal{H} \rightarrow \mathbb{R}^K$ that outputs mixture plausibility logits, and (iii) an advocacy-strength head $g_{\boldsymbol{\phi}_3} : \mathcal{H} \rightarrow \mathbb{R}^K$ that generates logits corresponding to class-specific advocacy strengths.

These logits are transformed into valid distributional parameters using appropriate activation functions. In particular, the base evidence and advocacy-strength logits are passed through exponential activations to yield positive parameters $\boldsymbol{\alpha}(\mathbf{x}_i)$ and $\boldsymbol{\tau}(\mathbf{x}_i)$, while a softmax function, denoted as $\sigma^{\mathrm{SM}}(\cdot)$, is applied to the mixture plausibility logits to produce mixture weight $\boldsymbol{\omega}(\mathbf{x}_i)$:

$$\boldsymbol{\alpha}(\mathbf{x}_i) = \exp(g_{\boldsymbol{\phi}_1}(\mathbf{z}_i)),$$
$$\boldsymbol{\omega}(\mathbf{x}_i) = \sigma^{\mathrm{SM}}(g_{\boldsymbol{\phi}_2}(\mathbf{z}_i)),$$
$$\boldsymbol{\tau}(\mathbf{x}_i) = \exp(g_{\boldsymbol{\phi}_3}(\mathbf{z}_i)),$$

so that $\boldsymbol{\alpha}(\mathbf{x}_i) \in \mathbb{R}_{>0}^K, \boldsymbol{\omega}(\mathbf{x}_i) \in \Delta^{K-1}$, and $\boldsymbol{\tau}(\mathbf{x}_i) \in \mathbb{R}_{>0}^K$.

To stabilize uncertainty estimation, we apply spectral normalization (Miyato et al., 2018) to the feature extractor $f_{\boldsymbol{\psi}}$ and the concentration head $g_{\boldsymbol{\phi}_1}$, which has been shown to

improve the robustness of UQ (Liu et al., 2023; Yoon & Kim, 2024).

Together, for each input $\mathbf{x}_i$, the predicted parameters $(\boldsymbol{\alpha}(\mathbf{x}_i), \boldsymbol{\omega}(\mathbf{x}_i), \boldsymbol{\tau}(\mathbf{x}_i))$ define a coherent probabilistic model over the latent class-probability vector $\boldsymbol{\pi}_i$ and the observed label $y_i$. We make this structure explicit by specifying the following generative process.

**Generative Process.** MoDEX defines the following generative model for each input $\mathbf{x}_i$:

$$L_i \sim \text{Cat}(\boldsymbol{\omega}(\mathbf{x}_i)),$$
$$\boldsymbol{\pi}_i \mid (L = k, \mathbf{x}_i) \sim \text{Dir}(\boldsymbol{\alpha}(\mathbf{x}_i) + \tau_k(\mathbf{x}_i)\mathbf{e}_k),$$
$$y_i \mid \boldsymbol{\pi}_i \sim \text{Cat}(\boldsymbol{\pi}_i).$$

The generative process provides an explicit interpretation of the courtroom-induced predictive distribution in Eq. (2). Specifically, a latent advocate index $L_i$ determines the class-specific opinion governing the latent class-probability vector $\boldsymbol{\pi}_i$, from which the observed $y_i$ is then drawn.

**Equivalence to the EFD Distribution.** The above generative model is equivalent to predicting an extended flexible Dirichlet (EFD) distribution (Ongaro et al., 2020) over $\boldsymbol{\pi}_i$,

$$\boldsymbol{\pi}_i \mid \mathbf{x}_i \sim \text{EFD}(\boldsymbol{\alpha}(\mathbf{x}_i), \boldsymbol{\omega}(\mathbf{x}_i), \boldsymbol{\tau}(\mathbf{x}_i)).$$

This links the courtroom-induced predictor in Eq. (2) to a well-defined parametric family with structured expressiveness on the simplex.

### 4.2. Objective Function

MoDEX is trained using a composite objective consisting of three terms. First, a mean squared error (MSE) loss is applied to the expected class probabilities, following standard EDL practice for uncertainty-aware predictions (Chen et al., 2025). Second, a Brier-score-based regularization is imposed on $\boldsymbol{\omega}(\mathbf{x}_i)$ to promote calibrated expert weighting and prevent degenerate solutions. Third, a KL-divergence-based regularization is applied to $\boldsymbol{\tau}(\mathbf{x}_i)$, providing a soft supervisory signal toward the correct class.

Given a labeled example $(\mathbf{x}_i, y_i)$ with a one-hot target $\mathbf{y}_i$, we define the training objective as

$$\mathcal{L}(\mathbf{x}_i, \mathbf{y}_i) = \underbrace{\|\mathbf{y}_i - \mathbb{E}_{\boldsymbol{\pi}_i \sim \text{EFD}(\boldsymbol{\alpha}(\mathbf{x}_i), \boldsymbol{\omega}(\mathbf{x}_i), \boldsymbol{\tau}(\mathbf{x}_i))}[\boldsymbol{\pi}_i]\|_2^2}_{\text{MSE on the predictive mean}}$$
$$+ \underbrace{\|\mathbf{y}_i - \boldsymbol{\omega}(\mathbf{x}_i)\|_2^2}_{\text{regularization for } \boldsymbol{\omega}} + \underbrace{D_{\text{KL}}(\sigma^{\text{SM}}(\boldsymbol{\tau}(\mathbf{x}_i)) \| \tilde{\mathbf{y}}_i)}_{\text{regularization for } \boldsymbol{\tau}},$$

where the smoothed target $\tilde{\mathbf{y}}_i$ is given by $\tilde{\mathbf{y}}_i = (1 - \epsilon)\mathbf{y}_i + \frac{\epsilon}{K-1}(\mathbf{1} - \mathbf{y}_i)$ with label-smoothing parameters $\epsilon \in [0, 1]$, and $\mathbf{1}$ is a vector of ones.

### 4.3. Prediction & Uncertainty Measures

**Prediction.** As a concrete instantiation of a second-order UQ framework, MoDEX specifies, for a test input $\mathbf{x}^\star$, a predictive distribution over the latent class-probability vector $\boldsymbol{\pi}^\star$:

$$\hat{p}(\boldsymbol{\pi}^\star \mid \mathbf{x}^\star) = \text{EFD}(\boldsymbol{\alpha}^\star(\mathbf{x}^\star), \boldsymbol{\omega}(\mathbf{x}^\star), \boldsymbol{\tau}(\mathbf{x}^\star)),$$

Marginalizing the latent class-probability vector $\boldsymbol{\pi}^\star$ yields the induced first-order predictive distribution over labels,

$$\hat{p}(y^\star = k \mid \mathbf{x}^\star) = \mathbb{E}_{\boldsymbol{\pi}^\star \sim \hat{p}(\boldsymbol{\pi}^\star \mid \mathbf{x}^\star)}[\pi_k], \quad \forall k \in [K].$$

Prediction is then performed by selecting the class with the highest expected class probability, i.e., $\hat{y}^\star = \text{argmax}_{k \in [K]} \hat{p}(y^\star = k \mid \mathbf{x}^\star)$.

This hierarchical construction induces two complementary notions of predictive uncertainty: (i) outcome-level uncertainty, reflecting ambiguity in the induced label distribution $\hat{p}(y^\star \mid \mathbf{x}^\star)$, and (ii) belief-level uncertainty, capturing dispersion in the second-order distribution $\hat{p}(\boldsymbol{\pi}^\star \mid \mathbf{x}^\star)$. In our framework, these are operationalized as aleatoric and epistemic uncertainty, respectively.

**Aleatoric Uncertainty.** We define aleatoric uncertainty as the entropy of the induced first-order predictive distribution over labels,

$$\text{AU}(\mathbf{x}^\star) \triangleq \mathbb{H}[\hat{p}(y^\star \mid \mathbf{x}^\star)] = -\sum_{k=1}^{K} \mu_k(\mathbf{x}^\star) \log \mu_k(\mathbf{x}^\star),$$

where $\mu_k(\mathbf{x}^\star) = \hat{p}(y^\star = k \mid \mathbf{x}^\star)$ denotes the predicted mean probability for class $k$.

This quantity measures the residual ambiguity in the predicted label after aggregating uncertainty in $\boldsymbol{\pi}^\star$ into a single outcome distribution. Under the courtroom analogy, AU captures the uncertainty in the final verdict, reflecting the ambiguity after deliberation consolidates advocates' beliefs.

**Epistemic Uncertainty.** We quantify epistemic uncertainty by the dispersion of $\boldsymbol{\pi}^\star$ under the second-order predictive distribution, measured via total variance:

$$\text{EU}(\mathbf{x}^\star) \triangleq \text{tr}\big(\text{Cov}_{\boldsymbol{\pi}^\star \sim \hat{p}(\boldsymbol{\pi}^\star \mid \mathbf{x}^\star)}[\boldsymbol{\pi}^\star]\big),$$

where $\text{Cov}_{\boldsymbol{\pi}^\star \sim \hat{p}(\boldsymbol{\pi}^\star \mid \mathbf{x}^\star)}[\boldsymbol{\pi}^\star]$ is the covariance matrix of $\boldsymbol{\pi}^\star$ under $\hat{p}(\boldsymbol{\pi}^\star \mid \mathbf{x}^\star)$ and $\text{tr}(\cdot)$ denotes the trace operator.

This quantity captures uncertainty over plausible class-probability vectors themselves. Under the courtroom analogy, EU reflects uncertainty prior to deliberation—i.e., the extent to which advocates' beliefs are imprecise or disagree with one another—encompassing both within-advocate uncertainty and across-advocate disagreement.

Closed-form moment expressions are deferred to Appendix A, algorithmic details to Appendix B, and the correspondence with the courtroom analogy to Appendix C.

# 5. Theoretical Advancements

MoDEX introduces key theoretical advancements that enhance both expressiveness and interpretability in UQ. First, we prove that MoDEX reduces to $\mathcal{F}$-EDL (Yoon & Kim, 2026) as a special case when $\tau_k(\mathbf{x}^\star) = \tau(\mathbf{x}^\star)$ for all $k \in [K]$, and further reduces to standard EDL (Sensoy et al., 2018) when $\tau(\mathbf{x}^\star) = 1$ and $\boldsymbol{\omega}(\mathbf{x}^\star) = \boldsymbol{\alpha}(\mathbf{x}^\star)/\|\boldsymbol{\alpha}(\mathbf{x}^\star)\|_1$ (Theorem 5.1). These reductions establish MoDEX as a generalization of prior EDL models, retaining their properties while adding flexibility for structured and input-adaptive uncertainty, in line with the courtroom analogy.

Second, we prove that the predictive distribution induced by MoDEX for $\mathbf{x}^\star$ admits two equivalent representations: (i) a weighted ensemble of $K$ Dirichlet (i.e., EDL) experts, each with expert-specific evidence $\boldsymbol{\alpha}_k(\mathbf{x}^\star)$, and (ii) a class- and input-dependent combination of an EDL predictor with base evidence $\boldsymbol{\alpha}(\mathbf{x}^\star)$ and a softmax predictor (Proposition 5.2, Theorem 5.3). These representations provide insights into how MoDEX combines different uncertainty modeling mechanisms across input regimes. For clean ID inputs, the predictive distribution is expected to be dominated by a single plausible expert in the mixture representation (Eq. 3) or by the base-evidence EDL predictor in the EDL-softmax decomposition (Eq. 4). In contrast, for ambiguous or OOD inputs, MoDEX may assign substantial weights to multiple mixture components in both representations, ensuring a more balanced aggregation of predictive contributions and, consequently, improved reliability and robustness in both prediction and UQ.

Third, we prove that the epistemic uncertainty induced by MoDEX for $\mathbf{x}^\star$ can be decomposed into two interpretable components: inter-expert uncertainty, which quantifies disagreement across experts, and intra-expert uncertainty, which reflects uncertainty within each expert due to insufficient or noisy evidence (Proposition 5.4). Together, these components illustrate how MoDEX enables rich modeling of epistemic uncertainty by capturing multiple sources of uncertainty.

**Theorem 5.1** (*MoDEX as a Generalization of EDL and $\mathcal{F}$-EDL*). *The class probability distribution induced by MoDEX for $\mathbf{x}^\star$ reduces to that of $\mathcal{F}$-EDL when $\tau_k(\mathbf{x}^\star) = \tau(\mathbf{x}^\star), \forall k \in [K]$, and further reduces to that of standard EDL when $\tau(\mathbf{x}^\star) = 1$ and $\boldsymbol{\omega}(\mathbf{x}^\star) = \boldsymbol{\alpha}(\mathbf{x}^\star)/\|\boldsymbol{\alpha}(\mathbf{x}^\star)\|_1$.*

**Proposition 5.2** (*MoDEX as a Mixture of EDL Experts*). *Let the predictive distribution induced by the $k$-th EDL expert for $\mathbf{x}^\star$ be defined as $\hat{p}_{\mathrm{EDL}}^{(k)}(y^\star \mid \mathbf{x}^\star) := \mathrm{Cat}\big(\boldsymbol{\mu}^{(k)}(\mathbf{x}^\star)\big)$, where $\boldsymbol{\mu}^{(k)}(\mathbf{x}^\star) := \mathbb{E}_{\boldsymbol{\pi}^\star \sim \mathrm{Dir}(\boldsymbol{\alpha}_k(\mathbf{x}^\star))}[\boldsymbol{\pi}^\star]$ is the mean class probability vector for the $k$-th expert. Then, the predictive distribution induced by MoDEX for $\mathbf{x}^\star$ is given by*

$$\hat{p}(y^\star \mid \mathbf{x}^\star) = \sum_{k=1}^{K} \omega_k(\mathbf{x}^\star) \times \hat{p}_{\mathrm{EDL}}^{(k)}(y^\star \mid \mathbf{x}^\star). \quad (3)$$

**Theorem 5.3** (*MoDEX as a Mixture of Base EDL and Softmax Predictors*). *Let the predictive distributions induced by the base EDL model and the softmax model for $\mathbf{x}^\star$ be: $\hat{p}_{\mathrm{EDL}}(y^\star \mid \mathbf{x}^\star) := \mathrm{Cat}(\frac{\boldsymbol{\alpha}(\mathbf{x}^\star)}{\|\boldsymbol{\alpha}(\mathbf{x}^\star)\|_1})$ and $\hat{p}_{\mathrm{SM}}(y^\star \mid \mathbf{x}^\star) := \mathrm{Cat}(\boldsymbol{\omega}(\mathbf{x}^\star))$. Then, the predictive distribution induced by MoDEX for $\mathbf{x}^\star$ is a mixture of the base EDL and softmax predictors:*

$$\begin{aligned} \hat{p}(y^\star \mid \mathbf{x}^\star) = &\, \lambda_{\mathrm{EDL}}(\mathbf{x}^\star)\,\hat{p}_{\mathrm{EDL}}(y^\star \mid \mathbf{x}^\star) \\ &+ \boldsymbol{\lambda}_{\mathrm{SM}}(\mathbf{x}^\star) \odot \hat{p}_{\mathrm{SM}}(y^\star \mid \mathbf{x}^\star), \end{aligned} \quad (4)$$

*where $\odot$ denotes the Hadamard (elementwise) product, and the mixture weights are:*

$$\lambda_{\mathrm{EDL}}(\mathbf{x}^\star) := \sum_{k=1}^{K} \frac{\|\boldsymbol{\alpha}(\mathbf{x}^\star)\|_1\,\omega_k(\mathbf{x}^\star)}{\|\boldsymbol{\alpha}(\mathbf{x}^\star)\|_1 + \tau_k(\mathbf{x}^\star)},$$

$$\boldsymbol{\lambda}_{\mathrm{SM}}(\mathbf{x}^\star) := \left[\frac{\tau_k(\mathbf{x}^\star)}{\|\boldsymbol{\alpha}(\mathbf{x}^\star)\|_1 + \tau_k(\mathbf{x}^\star)}\right]_{k=1}^{K}.$$

**Proposition 5.4** (*Epistemic Uncertainty Decomposition*). *Let the aggregated mean class probability vector be $\bar{\boldsymbol{\mu}}(\mathbf{x}^\star) := \sum_{k=1}^{K} \omega_k(\mathbf{x}^\star)\boldsymbol{\mu}^{(k)}(\mathbf{x}^\star)$. Then, the epistemic uncertainty induced by MoDEX for $\mathbf{x}^\star$ can be decomposed into inter-expert and intra-expert components, i.e., $\mathrm{EU}(\mathbf{x}^\star) = \mathrm{EU}_{\mathrm{inter}}(\mathbf{x}^\star) + \mathrm{EU}_{\mathrm{intra}}(\mathbf{x}^\star)$, where*

$$\mathrm{EU}_{\mathrm{inter}}(\mathbf{x}^\star) := \sum_{k=1}^{K} \omega_k(\mathbf{x}^\star)\left\|\boldsymbol{\mu}^{(k)}(\mathbf{x}^\star) - \bar{\boldsymbol{\mu}}(\mathbf{x}^\star)\right\|_2^2,$$

$$\mathrm{EU}_{\mathrm{intra}}(\mathbf{x}^\star) := \sum_{k=1}^{K} \omega_k(\mathbf{x}^\star) \sum_{j=1}^{K} \mathrm{Var}_{\boldsymbol{\pi}^\star \sim \mathrm{Dir}(\boldsymbol{\alpha}_k(\mathbf{x}^\star))}\big[\pi_j^\star\big].$$

Proofs of all theorems are provided in Appendix D.

# 6. Experiments

## 6.1. Overview of Experiments

We conducted extensive experiments to evaluate the UQ capability of MoDEX. First, we perform *quantitative* evaluations to assess the quality of the uncertainty measures induced by MoDEX across various UQ-related tasks and benchmarks (Section 6.2). Second, we present *qualitative* analysis demonstrating MoDEX's ability to represent uncertainty with interpretable semantics in practice (Section 6.3).

**Tasks.** We evaluate (i) classification accuracy on the ID test set; (ii) whether aleatoric uncertainty is higher for misclassified samples than for correct ones (misclassification detection) [1]; (iii) whether epistemic uncertainty is higher for OOD samples (OOD detection) [2]; and (iv) whether epistemic uncertainty increases for corrupted samples (distribution shift detection). We use aleatoric uncertainty for misclassification detection because it reflects ambiguity in

---
[1]labels: correct = 1, incorrect = 0
[2]labels: ID = 1, OOD = 0

*Table 1.* UQ-related downstream task results using CIFAR-10 and CIFAR-100 as the ID datasets. "Miscl. AUPR." denotes the AUPR scores for misclassification detection based on aleatoric uncertainty, and "SVHN | C-100" and "SVHN | TIN" indicate AUPR scores for OOD detection based on epistemic uncertainty, where the former corresponds to using CIFAR-10 as the ID dataset with SVHN and CIFAR-100 as the OOD datasets, and the latter corresponds to using CIFAR-100 as the ID dataset with SVHN and TIN as the OOD datasets. Baseline results are from the existing works (Chen et al., 2025; Yoon & Kim, 2026). Bold values indicate the best performance.

| Method | ID: CIFAR-10 | | | ID: CIFAR-100 | | |
| --- | --- | --- | --- | --- | --- | --- |
| | Test.Acc. | Miscl. AUPR. | SVHN \| C-100 | Test.Acc. | Miscl. AUPR. | SVHN \| TIN |
| Dropout | 90.16 $\pm$0.2 | 98.86 $\pm$0.6 | 78.40 $\pm$3.9 / 85.39 $\pm$0.6 | 65.94 $\pm$0.6 | 92.00 $\pm$0.3 | 71.83 $\pm$2.0 / 74.93 $\pm$0.6 |
| EDL | 88.48 $\pm$0.3 | 98.74 $\pm$0.7 | 82.32 $\pm$1.2 / 87.13 $\pm$0.3 | 45.91 $\pm$5.6 | 91.28 $\pm$0.8 | 56.21 $\pm$3.1 / 70.13 $\pm$2.0 |
| $\mathcal{I}$-EDL | 89.20 $\pm$0.3 | 98.72 $\pm$0.1 | 82.96 $\pm$2.2 / 84.84 $\pm$0.6 | 66.38 $\pm$0.5 | 92.84 $\pm$0.1 | 67.51 $\pm$2.9 / 75.86 $\pm$0.3 |
| R-EDL | 90.09 $\pm$0.3 | 98.98 $\pm$0.1 | 85.00 $\pm$1.2 / 87.73 $\pm$0.3 | 63.53 $\pm$0.5 | 92.69 $\pm$0.2 | 61.80 $\pm$3.4 / 69.78 $\pm$1.3 |
| DAEDL | 91.11 $\pm$0.2 | 99.08 $\pm$0.0 | 85.54 $\pm$1.4 / 88.19 $\pm$0.1 | 66.01 $\pm$2.6 | 86.00 $\pm$0.3 | 72.07 $\pm$4.1 / 77.40 $\pm$1.6 |
| Re-EDL | 90.09 $\pm$0.3 | 98.81 $\pm$0.1 | 89.94 $\pm$1.4 / 88.31 $\pm$0.2 | 64.53 $\pm$0.7 | 92.63 $\pm$0.4 | 68.37 $\pm$2.8 / 76.87 $\pm$0.7 |
| $\mathcal{F}$-EDL | 91.19 $\pm$0.2 | 99.10 $\pm$0.0 | 91.20 $\pm$1.3 / 88.37 $\pm$0.3 | 69.40 $\pm$0.2 | 94.01 $\pm$0.1 | 75.35$\pm$2.3 / 80.58 $\pm$0.2 |
| **MoDEX** | **92.46** $\pm$**0.2** | **99.18** $\pm$**0.0** | **91.58** $\pm$**0.4** / **89.28** $\pm$**0.3** | **75.91** $\pm$**0.9** | **96.17** $\pm$**0.2** | **77.90** $\pm$**1.5** / **81.76** $\pm$**0.3** |

*Table 2.* AUPR scores for distribution shift detection from CIFAR-10 to CIFAR-10-C based on epistemic uncertainty estimates. Results are reported for selected corruption severity levels $\mathcal{C} \in \{1, 3, 5\}$, averaged across 19 corruption types.

| Method | $\mathcal{C} = 1$ | $\mathcal{C} = 3$ | $\mathcal{C} = 5$ |
| --- | --- | --- | --- |
| MSP | 56.39 $\pm$0.7 | 65.86 $\pm$1.3 | 75.01 $\pm$1.8 |
| EDL | 55.56 $\pm$0.7 | 63.38 $\pm$1.4 | 73.12 $\pm$1.3 |
| $\mathcal{I}$-EDL | 56.35 $\pm$0.5 | 65.23 $\pm$1.2 | 74.91 $\pm$1.9 |
| R-EDL | 57.17 $\pm$0.5 | 67.33 $\pm$1.2 | 76.80 $\pm$1.2 |
| DAEDL | 55.90 $\pm$0.8 | 63.69 $\pm$1.4 | 73.45 $\pm$1.5 |
| Re-EDL | 56.93 $\pm$0.6 | 66.84 $\pm$1.2 | 76.05 $\pm$1.5 |
| $\mathcal{F}$-EDL | 58.16 $\pm$0.5 | 68.44 $\pm$0.8 | 78.52 $\pm$1.1 |
| **MoDEX** | **58.72** $\pm$**0.5** | **70.24** $\pm$**0.5** | **80.63** $\pm$**0.5** |

*Table 3.* UQ-related downstream task results on CIFAR-10-LT under severe class imbalance ($\rho = 0.01$).

| Method | Test.Acc. | Miscl. AUPR. | SVHN \| C-100 |
| --- | --- | --- | --- |
| Dropout | 39.22 $\pm$3.1 | 63.62 $\pm$2.7 | 33.33 $\pm$1.7 / 54.17 $\pm$1.1 |
| EDL | 42.62 $\pm$2.7 | 82.63 $\pm$1.7 | 51.99 $\pm$3.8 / 66.86 $\pm$0.9 |
| $\mathcal{I}$-EDL | 57.88 $\pm$1.3 | 84.10 $\pm$1.3 | 52.85 $\pm$6.8 / 69.19 $\pm$1.3 |
| R-EDL | 63.36 $\pm$1.0 | 78.34 $\pm$1.0 | 48.71 $\pm$7.1 / 64.20 $\pm$1.4 |
| DAEDL | 63.36 $\pm$1.4 | 82.15 $\pm$1.0 | 51.03 $\pm$5.6 / 65.31 $\pm$1.2 |
| Re-EDL | 66.27 $\pm$0.4 | 83.40 $\pm$1.2 | 39.48 $\pm$9.4 / 62.51 $\pm$2.4 |
| $\mathcal{F}$-EDL | 63.73 $\pm$1.4 | 85.99 $\pm$1.7 | 62.56 $\pm$2.8 / 70.18 $\pm$2.0 |
| **MoDEX** | **71.53** $\pm$**1.1** | **91.90** $\pm$**1.2** | **72.05** $\pm$**3.5** / **76.52** $\pm$**0.9** |

the final predictive distribution, whereas epistemic uncertainty is used for OOD and distribution-shift detection to capture model uncertainty under unfamiliar or shifted inputs. Detection is based on negative uncertainty measures, with performance evaluated by the area under the precision-recall curve (AUPR), normalized to 0-100.

**ID Settings & Datasets.** We evaluate the model on benchmarks covering three ID settings: (i) standard, (ii) long-tailed, and (iii) simultaneous long-tailed and noisy. For the standard setting, we use CIFAR-10 and CIFAR-100 as ID datasets. In the long-tailed setting, we adopt CIFAR-10-LT (Cui et al., 2019), an imbalanced variant of CIFAR-10, as the ID dataset. To simulate a simultaneous long-tailed

and noisy setting, we introduce class imbalance into Dirty-MNIST (DMNIST) (Mukhoti et al., 2023), a noisy version of MNIST with ambiguous samples, resulting in its imbalanced version termed DMNIST-LT. For distribution shift detection, we employ CIFAR-10-C (Hendrycks & Dietterich, 2019), which applies a variety of corruption-based shifts to CIFAR-10. For OOD detection, SVHN and CIFAR-100 are used as OOD datasets for CIFAR-10 and CIFAR-10-LT, while SVHN and TinyImageNet-200 are used for CIFAR-100. FMNIST is used as the OOD dataset for DMNIST-LT.

**Baselines.** We compared MoDEX against state-of-the-art EDL approaches, including EDL (Sensoy et al., 2018) and its extensions: $\mathcal{I}$-EDL (Deng et al., 2023), R-EDL (Chen et al., 2024), DAEDL (Yoon & Kim, 2024), Re-EDL (Chen et al., 2025), and $\mathcal{F}$-EDL (Yoon & Kim, 2026). Additionally, we included MC Dropout (Gal & Ghahramani, 2016) as a classical Bayesian UQ baseline, and MSP (Hendrycks & Gimpel, 2017) as a baseline for distribution shift detection.

**Implementations.** For DMNIST-LT, we employ a lightweight CNN with three convolutional layers followed by three dense layers. For CIFAR-10 and CIFAR-10-LT, we use VGG-16, while ResNet-18 is used for CIFAR-100. To estimate additional courtroom parameters $\omega$ and $\tau$, we add two shallow MLP heads with one to two layers each, introducing a minimal overhead (e.g., 0.9% for CIFAR-10).

Experimental details are in Appendix E, with additional results in Appendix F. Our implementation is publicly available at https://github.com/TaeseongYoon/MoDEX.

### 6.2. Quantitative Experiments

**Standard Setting.** We evaluate MoDEX under the standard setting using CIFAR-10 and CIFAR-100 as the ID datasets, with clean and class-balanced training data. As shown in Table 1, MoDEX achieves state-of-the-art performance across all tasks, with a notable improvement in classification accuracy. This improvement stems from MoDEX's uncertainty-

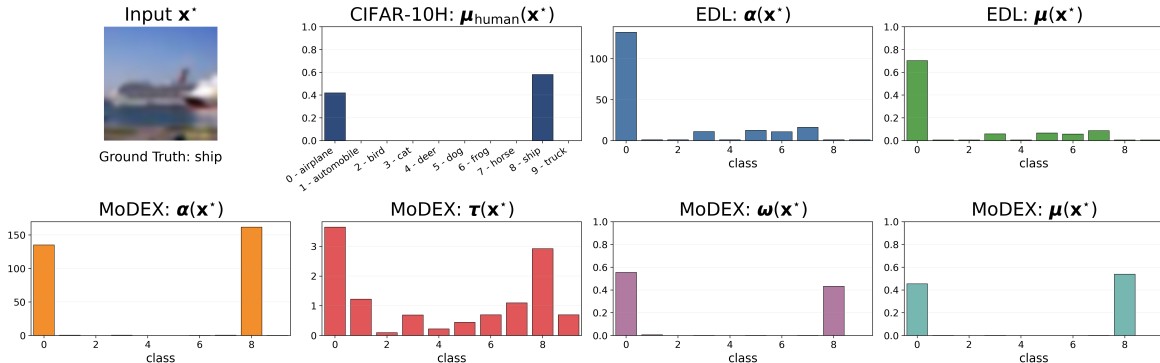

*Figure 1.* Uncertainty-aware classification results for EDL and MoDEX, both trained on CIFAR-10-LT and evaluated on test input $\mathbf{x}^\star$ with the ground-truth label *ship* (tail class) and semantic ambiguity with *airplane* (head class). The top row shows the input image, the human-annotated class distribution from CIFAR-10H, and EDL's concentration parameters $\boldsymbol{\alpha}(\mathbf{x}^\star)$ and predicted mean $\boldsymbol{\mu}(\mathbf{x}^\star)$. The bottom row shows the courtroom parameters induced by MoDEX, along with the predicted mean $\boldsymbol{\mu}(\mathbf{x}^\star)$.

aware classification mechanism, based on the courtroom analogy, which contrasts with the expressivity-driven modifications often employed by other EDL variants that may enhance UQ but can adversely affect classification accuracy. Additionally, Table 2 shows that MoDEX excels at detecting distribution shifts, likely due to its explicit modeling of uncertainty arising from insufficient evidence (Proposition 5.4).

**Long-Tailed Setting.** We evaluate the robustness of MoDEX under class imbalance on CIFAR-10-LT with imbalance factors $\rho \in \{0.1, 0.01\}$. Reliable UQ becomes particularly challenging in long-tailed regimes, as head-class dominance often leads to miscalibrated uncertainty estimates for tail-class samples. This effect blurs the distinction between tail ID inputs and OOD samples, thereby degrading OOD detection performance. As shown in Table 3, MoDEX consistently outperforms all competing baselines under severe class imbalance ($\rho = 0.01$), achieving especially strong improvements in OOD detection. [3] This robustness is attributed to MoDEX's structured uncertainty modeling, which mitigates head-class bias by assigning a dedicated Dirichlet expert to each class (Proposition 5.2) and by learning input- and class-dependent mixture coefficients $\boldsymbol{\lambda}_{\mathrm{SM}}(\mathbf{x}^\star)$ that adaptively balance the base EDL and softmax predictors in a class-wise manner (Theorem 5.3).

**Long-Tailed and Noisy Setting.** We further evaluate MoDEX under a more challenging scenario where the dataset is both noisy and class-imbalanced. This setting presents a key challenge: epistemic uncertainty must distinguish noisy tail-class examples—which are inherently ambiguous—from genuine OOD samples. As shown in Table 4, MoDEX demonstrates strong UQ performance under these challenging conditions. This robustness is attributed to its enhanced expressivity, which, through a principled gen-

---

[3]MoDEX also performs favorably under mild class imbalance ($\rho = 0.1$); detailed results are in Table 14 in Appendix.

*Table 4.* UQ-related downstream task results using DMNIST-LT ($\rho = 0.01$) as the ID dataset.

| Method | Test.Acc. | Miscl. AUPR. | FMNIST |
|---|---|---|---|
| EDL | $38.72 \pm 8.6$ | $87.03 \pm 2.4$ | $91.30 \pm 1.9$ |
| $\mathcal{I}$-EDL | $59.69 \pm 7.6$ | $86.25 \pm 1.1$ | $95.75 \pm 0.9$ |
| R-EDL | $66.65 \pm 0.8$ | $86.74 \pm 0.8$ | $95.33 \pm 1.1$ |
| DAEDL | $66.11 \pm 1.4$ | $86.47 \pm 0.5$ | $98.06 \pm 0.6$ |
| Re-EDL | $64.91 \pm 4.5$ | $87.81 \pm 6.2$ | $93.21 \pm 4.2$ |
| $\mathcal{F}$-EDL | $67.20 \pm 1.2$ | $86.20 \pm 1.5$ | $96.84 \pm 0.4$ |
| **MoDEX** | **69.42** $\pm 1.3$ | **88.76** $\pm 1.0$ | **98.88** $\pm 0.2$ |

eralization of prior methods, enables MoDEX to effectively handle realistic and unforeseen challenges (Theorem 5.1).

### 6.3. Qualitative Evaluation: Semantically Interpretable, Uncertainty-Aware Classification

We show that MoDEX enables semantically interpretable, uncertainty-aware classification in challenging settings. Figure 1 compares EDL and MoDEX trained on CIFAR-10-LT ($\rho = 0.01$) and evaluated on an ambiguous CIFAR-10 test example $\mathbf{x}^\star$. The ground-truth label is *ship* (class 8, a tail class), yet the image is semantically confusable with *airplane* (class 0, a head class), as reflected in the CIFAR-10H human-annotated distribution $\boldsymbol{\mu}_{\mathrm{human}}(\mathbf{x}^\star)$ (Peterson et al., 2019).

Under severe class imbalance, EDL collapses its evidence toward *airplane*, yielding an overconfident and incorrect prediction. In contrast, MoDEX offers an interpretable aggregation of uncertainty via its courtroom-inspired formulation. Using the decomposition $\alpha_k(\mathbf{x}^\star) = \boldsymbol{\alpha}(\mathbf{x}^\star) + \tau_k(\mathbf{x}^\star)\mathbf{e}_k$, MoDEX separates *base evidence* $\boldsymbol{\alpha}(\mathbf{x}^\star)$ from *class-specific advocacy strength* $\tau_k(\mathbf{x}^\star), \forall k \in [K]$, thereby preventing head-class bias from being entangled with the shared evidence. Here, $\boldsymbol{\alpha}(\mathbf{x}^\star)$ is appropriately biased toward the tail class and favors *ship*, based on objective cues (e.g., sea texture and ship-like silhouette). Meanwhile, $\boldsymbol{\tau}(\mathbf{x}^\star)$ is amplified for *airplane*, suggesting that the advocate for *airplane* can present a stronger argument, owing to its greater

exposure to visually similar patterns during training. Furthermore, $\omega(\mathbf{x}^\star)$ assigns substantial plausibility to both *ship* and *airplane*, reflecting that both sides present strong arguments. Aggregating these components yields a predictive mean $\mu(\mathbf{x}^\star)$ that assigns non-negligible probability on *airplane* while correctly favoring the tail class *ship*, closely aligning with $\mu_{\mathrm{human}}(\mathbf{x}^\star)$.

Additional qualitative evaluation is provided in Appendix G.

# 7. Conclusion

**Summary.** We introduced a courtroom analogy that provides a structured and interpretable perspective on uncertainty-aware classification. Building on this perspective, we proposed MoDEX, a single-pass neural framework that realizes uncertainty aggregation as a structured mixture of Dirichlet experts with explicit semantic roles. Supported by theoretical foundations, MoDEX exhibits interpretable uncertainty behavior and achieves state-of-the-art performance across a wide range of UQ-related downstream tasks and diverse benchmarks.

**Limitations.** Despite its effectiveness, several limitations remain. First, the framework is currently limited to classification; extending the courtroom analogy to uncertainty-aware regression, for example, by building on deep evidential regression (Amini et al., 2020) is a natural next step. Second, like other single-pass UQ approaches, MoDEX lacks explicit supervision on epistemic, or second-order, uncertainty (Bengs et al., 2022; 2023; Juergens et al., 2024; Shen et al., 2024). Thus, its epistemic uncertainty should not be interpreted as a uniquely recovered ground-truth second-order predictive distribution from input-label supervision alone. Instead, it is better understood as a structured, model-induced uncertainty signal arising from the courtroom-based aggregation mechanism, similar in spirit to deterministic UQ methods (Van Amersfoort et al., 2020; Liu et al., 2023), where uncertainty is induced by modeling assumptions and inductive biases.

**Future Directions.** Beyond these limitations, MoDEX also opens several directions for future work. First, it could be further tailored to challenging ID regimes such as long-tailed classification. In particular, its class-wise advocate structure could be combined with rebalancing or logit-adjustment strategies (Menon et al., 2020; Park et al., 2024; Lee et al., 2025) to further improve calibration under severe class imbalance. Second, the model-induced epistemic uncertainty of MoDEX could be further strengthened by incorporating Bayesian or ensemble-based guidance (Lakshminarayanan et al., 2017; Malinin et al., 2019; Wang & Ji, 2024), or by adopting ambiguity-inducing training schemes such as mixup (Zhang et al., 2018) while preserving the courtroom structure. Third, the framework offers a flexible foundation for broader EDL applications, including domain adaptation, where distribution shifts can be modeled as changes in the influence of learned Dirichlet experts, analogous to reweighting precedents in a courtroom.

# Impact Statement

Reliable UQ is essential for the safe and trustworthy deployment of machine learning systems, particularly in high-stakes domains such as healthcare, finance, and manufacturing (Seoni et al., 2023; Blasco et al., 2024; Choi et al., 2024). While many existing UQ approaches emphasize expressive uncertainty representations or performance on UQ-related downstream tasks, they often provide limited insight into how predictive uncertainty is structured and aggregated. This lack of transparency can hinder interpretability and responsible decision-making. This work introduces the courtroom analogy as a principled framework for uncertainty-aware classification and proposes MoDEX, a single-pass neural architecture that instantiates this perspective through a structured mixture of Dirichlet experts. MoDEX produces semantically interpretable uncertainty estimates while remaining computationally efficient, and it demonstrates strong and consistent performance across a wide range of UQ-related downstream tasks. At the same time, MoDEX is a data-driven uncertainty estimation method whose uncertainty estimates are learned from data under specific modeling assumptions. Consequently, its reliability depends on the suitability of the Dirichlet mixture formulation and the coverage of the training data. As a result, MoDEX may not fully capture all sources of uncertainty in highly complex, ambiguous, or adversarial environments. Accordingly, MoDEX should be applied with appropriate caution in practice, with its uncertainty behavior validated under realistic deployment conditions and interpreted in conjunction with domain expertise and human oversight.

# Acknowledgements

This work was supported by the National Research Foundation of Korea (NRF) grant funded by the Korea government (MSIT) (2023R1A2C2005453, RS-2023-00218913).

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

# Appendix for the Paper

## "Courtroom Analogy: New Perspective on Uncertainty-Aware Classification"

## A. First and Second-Order Moments of MoDEX under Courtroom and EFD Formulations.

In this section, we derive the first- and second-order moments of the latent class probability vector $\boldsymbol{\pi}^\star$ for input $\mathbf{x}^\star$, induced by MoDEX under two equivalent formulations: the structured mixture of Dirichlet (courtroom formulation) and the extended flexible Dirichlet (EFD) formulation. These moments are essential for computing the objective functions and uncertainty measures used in our paper in closed form. We begin by briefly explaining each formulation, followed by the closed-form derivations of the first- and second-order moments.

### A.1. Courtroom Formulation

The class probability distribution for the test input $\mathbf{x}^\star$ induced by MoDEX under the courtroom analogy is given by:

$$p(\boldsymbol{\pi}^\star \mid \mathbf{x}^\star) = \sum_{k=1}^{K} \omega_k(\mathbf{x}^\star) \operatorname{Dir}(\boldsymbol{\pi}^\star \mid \boldsymbol{\alpha}_k(\mathbf{x}^\star))$$

where $\boldsymbol{\alpha}_k(\mathbf{x}^\star) := \boldsymbol{\alpha}(\mathbf{x}^\star) + \tau_k(\mathbf{x}^\star)\mathbf{e}_k$. Here, $\mathbf{e}_k$ is the $k$-th standard basis vector, $\omega_k(\mathbf{x}^\star) > 0$, and weights satisfy the constraint $\sum_{k=1}^{K} \omega_k(\mathbf{x}^\star) = 1$.

### A.2. EFD Formulation

**Definition of the EFD distribution.** The extended flexible Dirichlet (EFD) distribution (Ongaro et al., 2020) generalizes both the Dirichlet and flexible Dirichlet distributions (Ongaro & Migliorati, 2013). The EFD distribution is generated by normalizing a suitable basis defined as:

$$Y_k = W_k + Z_k U_k, \quad \forall k \in [K],$$

where $W_k \sim \operatorname{Ga}(\alpha_k, \beta)$, $U_k \sim \operatorname{Ga}(\tau_k, \beta)$ are independent Gamma random variables with same scale parameter $\beta$. Additionally, $\mathbf{Z} \sim \operatorname{Mu}(1, \boldsymbol{\omega})$ is a multinomial random variable independent of $U_k$'s and the $W_k$'s. The EFD distribution, denoted as $\boldsymbol{\pi} \sim \operatorname{EFD}(\boldsymbol{\alpha}, \boldsymbol{\omega}, \boldsymbol{\tau})$, is the distribution of normalized vector $\boldsymbol{\pi} = (\pi_1, \ldots, \pi_K)$, where

$$\pi_k = \frac{Y_k}{\sum_{k=1}^{K} Y_k}, \quad \forall k \in [K].$$

**EFD Formulation.** As discussed in the main text, the class probability distribution induced by MoDEX is equivalent to the EFD distribution. Specifically, for $\mathbf{x}^\star$, the class probability distribution can be written as:

$$p(\boldsymbol{\pi}^\star \mid \mathbf{x}^\star) = \operatorname{EFD}(\boldsymbol{\pi}^\star \mid \boldsymbol{\alpha}(\mathbf{x}^\star), \boldsymbol{\omega}(\mathbf{x}^\star), \boldsymbol{\tau}(\mathbf{x}^\star)).$$

### A.3. First and Second-Order Moments

The first and second-order moments of the class probability vector $\boldsymbol{\pi}^\star$ can be derived under both formulations. The closed-form expressions for the mean and variance are provided below.

**Mean.** The mean under the courtroom formulation can be derived from the courtroom formulation using the linearity of expectation and the closed-form expectations for the Dirichlet distribution:

$$\begin{aligned}
\mathbb{E}_{\boldsymbol{\pi}^\star}[\pi_k^\star \mid \mathbf{x}^\star] &= \sum_{j=1}^{K} \omega_j(\mathbf{x}^\star) \times \mathbb{E}_{\boldsymbol{\pi}^\star \sim \operatorname{Dir}(\boldsymbol{\alpha}(\mathbf{x}^\star) + \tau_j(\mathbf{x}^\star)\mathbf{e}_j)}[\pi_k^\star] \\
&= \sum_{j=1}^{K} \omega_j(\mathbf{x}^\star) \times \left( \frac{\alpha_k(\mathbf{x}^\star) + \tau_j(\mathbf{x}^\star)\mathbb{I}\{j = k\}}{\|\boldsymbol{\alpha}(\mathbf{x}^\star)\|_1 + \tau_j(\mathbf{x}^\star)} \right) \\
&= \alpha_k(\mathbf{x}^\star)\kappa_1(\mathbf{x}^\star) + \tau_k(\mathbf{x}^\star)\frac{\omega_k(\mathbf{x}^\star)}{\|\boldsymbol{\alpha}(\mathbf{x}^\star)\|_1 + \tau_k(\mathbf{x}^\star)},
\end{aligned}$$

where

$$\kappa_1(\mathbf{x}^\star) = \sum_{j=1}^{K} \frac{\omega_j(\mathbf{x}^\star)}{\|\boldsymbol{\alpha}(\mathbf{x}^\star)\|_1 + \tau_j(\mathbf{x}^\star)}$$

This expression corresponds to the mean of the EFD distribution as described in Ongaro et al. (2020).

**Variance.** The variance for the EFD distribution, as presented in Ongaro et al. (2020), is given by:

$$\mathrm{Var}_{\boldsymbol{\pi}^\star}(\pi_k^\star \mid \mathbf{x}^\star) = \alpha_k(\mathbf{x}^\star)^2 \left(\kappa_2(\mathbf{x}^\star) - \kappa_1(\mathbf{x}^\star)^2\right) + \frac{\omega_k(\mathbf{x}^\star)\,\tau_k(\mathbf{x}^\star)\,(2\alpha_k(\mathbf{x}^\star) + \tau_k(\mathbf{x}^\star) + 1)}{(\|\boldsymbol{\alpha}(\mathbf{x}^\star)\|_1 + \tau_k(\mathbf{x}^\star))(\|\boldsymbol{\alpha}(\mathbf{x}^\star)\|_1 + \tau_k(\mathbf{x}^\star) + 1)}$$
$$+ \alpha_k(\mathbf{x}^\star)\kappa_2(\mathbf{x}^\star) - \frac{\omega_k(\mathbf{x}^\star)^2 \tau_k(\mathbf{x}^\star)^2}{(\|\boldsymbol{\alpha}(\mathbf{x}^\star)\|_1 + \tau_k(\mathbf{x}^\star))^2} - \kappa_1(\mathbf{x}^\star) \frac{2\alpha_k(\mathbf{x}^\star)\omega_k(\mathbf{x}^\star)\tau_k(\mathbf{x}^\star)}{\|\boldsymbol{\alpha}(\mathbf{x}^\star)\|_1 + \tau_k(\mathbf{x}^\star)},$$

where

$$\kappa_1(\mathbf{x}^\star) = \sum_{k=1}^{K} \frac{\omega_k(\mathbf{x}^\star)}{\|\boldsymbol{\alpha}(\mathbf{x}^\star)\|_1 + \tau_k(\mathbf{x}^\star)}, \quad \kappa_2(\mathbf{x}^\star) = \sum_{k=1}^{K} \frac{\omega_k(\mathbf{x}^\star)}{(\|\boldsymbol{\alpha}(\mathbf{x}^\star)\|_1 + \tau_k(\mathbf{x}^\star))(\|\boldsymbol{\alpha}(\mathbf{x}^\star)\|_1 + \tau_k(\mathbf{x}^\star) + 1)}$$

Furthermore, an equivalent but differently represented closed-form expression for the EFD distribution can be derived using the courtroom formulation. For a detailed derivation, refer to the proof of Proposition 5.4 in Appendix D.

## B. Algorithm

We summarize the algorithms for MoDEX as follows. Algorithm 1 outlines the training procedure, while Algorithm 2 describes the prediction and UQ process. Both algorithms are straightforward to implement, demonstrating the practical applicability of MoDEX.

---

**Algorithm 1** MoDEX Training

---

1: **Input:** Training data $\mathcal{D}_{\mathrm{tr}} = \{(\mathbf{x}_i, \mathbf{y}_i)\}_{i=1}^{N}$ with one-hot $\mathbf{y}_i \in \{0,1\}^K$; feature extractor $f_{\boldsymbol{\psi}}$; heads $g_{\boldsymbol{\phi}_1}, g_{\boldsymbol{\phi}_2}, g_{\boldsymbol{\phi}_3}$; max epochs $T_{\max}$; learning rate $\eta$; label smoothing $\epsilon \in [0,1]$.
2: **Output:** Trained parameters $\{\hat{\boldsymbol{\psi}}, \hat{\boldsymbol{\phi}}_1, \hat{\boldsymbol{\phi}}_2, \hat{\boldsymbol{\phi}}_3\}$.
3: **for** $t = 1$ **to** $T_{\max}$ **do**
4:    **for each** mini-batch $\mathcal{B} \subset \mathcal{D}_{\mathrm{tr}}$ **do**
5:       **for each** $(\mathbf{x}_i, \mathbf{y}_i) \in \mathcal{B}$ **do**
6:          **Step 1 (Compute courtroom parameters):**
7:             $\mathbf{z}_i \leftarrow f_{\boldsymbol{\psi}}(\mathbf{x}_i)$
8:             $\boldsymbol{\alpha}(\mathbf{x}_i) \leftarrow \exp(g_{\boldsymbol{\phi}_1}(\mathbf{z}_i))$
9:             $\boldsymbol{\omega}(\mathbf{x}_i) \leftarrow \sigma^{\mathrm{SM}}(g_{\boldsymbol{\phi}_2}(\mathbf{z}_i))$
10:            $\boldsymbol{\tau}(\mathbf{x}_i) \leftarrow \exp(g_{\boldsymbol{\phi}_3}(\mathbf{z}_i))$
11:          **Step 2 (Compute predictive mean):**
12:            $\boldsymbol{\mu}(\mathbf{x}_i) \leftarrow \mathbb{E}_{\boldsymbol{\pi}_i}[\boldsymbol{\pi}_i \mid \mathbf{x}_i]$
13:          **Step 3 (Compute objective):**
14:            $\tilde{\mathbf{y}}_i \leftarrow (1-\epsilon)\mathbf{y}_i + \frac{\epsilon}{K-1}(\mathbf{1} - \mathbf{y}_i)$
15:            $\ell(\mathbf{x}_i, \mathbf{y}_i) \leftarrow \|\mathbf{y}_i - \boldsymbol{\mu}(\mathbf{x}_i)\|_2^2 + \|\mathbf{y}_i - \boldsymbol{\omega}(\mathbf{x}_i)\|_2^2 + D_{\mathrm{KL}}\big(\sigma^{\mathrm{SM}}(\boldsymbol{\tau}(\mathbf{x}_i)) \,\|\, \tilde{\mathbf{y}}_i\big)$
16:       **end for**
17:       $\mathcal{L} \leftarrow \frac{1}{|\mathcal{B}|} \sum_{(\mathbf{x}_i, \mathbf{y}_i) \in \mathcal{B}} \ell(\mathbf{x}_i, \mathbf{y}_i)$
18:       **Adam update:** $\{\boldsymbol{\psi}, \boldsymbol{\phi}_1, \boldsymbol{\phi}_2, \boldsymbol{\phi}_3\} \leftarrow \mathrm{Adam}(\nabla\mathcal{L}, \eta)$
19:       **Spectral normalization:** apply $\mathrm{SpectNorm}(\cdot)$ to $f_{\boldsymbol{\psi}}$ and $g_{\boldsymbol{\phi}_1}$
20:    **end for**
21: **end for**
22: **Return:** $\{\hat{\boldsymbol{\psi}}, \hat{\boldsymbol{\phi}}_1, \hat{\boldsymbol{\phi}}_2, \hat{\boldsymbol{\phi}}_3\}$

---

---

**Algorithm 2** MoDEX Prediction and Uncertainty Quantification

---

1: **Input:** Test input $\mathbf{x}^\star$; trained parameters $\{\hat{\psi}, \hat{\phi}_1, \hat{\phi}_2, \hat{\phi}_3\}$
2: **Output:** Predicted label $\hat{y}$; predictive mean $\boldsymbol{\mu}^\star$; aleatoric uncertainty AU; epistemic uncertainty EU.
3: **Step 1 (Compute courtroom parameters):**
4:   $\mathbf{z}^\star \leftarrow f_{\hat{\psi}}(\mathbf{x}^\star)$
5:   $\boldsymbol{\alpha}(\mathbf{x}^\star) \leftarrow \exp(g_{\hat{\phi}_1}(\mathbf{z}^\star))$
6:   $\boldsymbol{\omega}(\mathbf{x}^\star) \leftarrow \sigma^{\mathrm{SM}}(g_{\hat{\phi}_2}(\mathbf{z}^\star))$
7:   $\boldsymbol{\tau}(\mathbf{x}^\star) \leftarrow \exp(g_{\hat{\phi}_3}(\mathbf{z}^\star))$
8: **Step 2 (Prediction):**
9:   $\boldsymbol{\mu}(\mathbf{x}^\star) \leftarrow \mathbb{E}_{\boldsymbol{\pi}^\star}[\boldsymbol{\pi}^\star \mid \mathbf{x}^\star]$
10:   $\hat{y}^\star \leftarrow \arg\max_{k \in [K]} \mu_k(\mathbf{x}^\star)$
11: **Step 3 (Uncertainty quantification):**
12:   $\mathrm{AU}(\mathbf{x}^\star) \leftarrow -\sum_{k=1}^{K} \mu_k(\mathbf{x}^\star) \log \mu_k(\mathbf{x}^\star)$
13:   $\mathrm{EU}(\mathbf{x}^\star) \leftarrow \mathrm{tr}(\mathrm{Cov}_{\boldsymbol{\pi}^\star}[\boldsymbol{\pi}^\star])$
14: **Return:** $\hat{y}^\star, \boldsymbol{\mu}(\mathbf{x}^\star), \mathrm{AU}(\mathbf{x}^\star), \mathrm{EU}(\mathbf{x}^\star)$

---

## C. Correspondence between the Courtroom Analogy and the MoDEX Components

This section formalizes the correspondence between the key concepts of the courtroom analogy and their neural instantiation in MoDEX. The goal is to clarify how the abstract notions introduced in the main text—such as advocates, evidence, advocacy, and verdict—are concretely realized within the MoDEX formulation. We organize this correspondence into two parts. First, we describe how general courtroom concepts correspond to the core components of MoDEX, including the input, Dirichlet experts, and their associated parameters. Second, we explain how beliefs, predictions, and uncertainty measures in the courtroom analogy map to predictive and uncertainty quantities in MoDEX. These correspondences are summarized in Table 5 and Table 6 and elaborated below.

*Table 5.* Correspondence between the general terms in the courtroom analogy and the corresponding components in MoDEX.

| Courtroom Concept | MoDEX |
|---|---|
| Case | Input $\mathbf{x}_i$ |
| Advocate $k$ | Dirichlet expert $k$ |
| Advocate $k$'s opinion | $\mathrm{Dir}(\boldsymbol{\alpha}_k(\mathbf{x}_i))$ |
| Shared evidence | $\boldsymbol{\alpha}(\mathbf{x}_i)$ |
| Class-specific advocacy | $\boldsymbol{\tau}(\mathbf{x}_i)$ |
| Advocacy plausibility | $\boldsymbol{\omega}(\mathbf{x}_i)$ |

*Table 6.* Correspondence between prediction and uncertainty measures in MoDEX and their interpretation under the courtroom analogy.

| Courtroom Concept | MoDEX |
|---|---|
| Belief distribution | $\hat{p}(\boldsymbol{\pi}^\star \mid \mathbf{x}^\star)$ |
| Final verdict distribution | $\hat{p}(y^\star \mid \mathbf{x}^\star)$ |
| Final verdict | $\hat{y}(\mathbf{x}^\star)$ |
| Residual verdict ambiguity | $\mathrm{AU}(\mathbf{x}^\star)$ |
| Pre-deliberation belief uncertainty | $\mathrm{EU}(\mathbf{x}^\star)$ |
| Inter-advocate disagreement | $\mathrm{EU}_{\mathrm{inter}}(\mathbf{x}^\star)$ |
| Intra-advocate imprecision | $\mathrm{EU}_{\mathrm{intra}}(\mathbf{x}^\star)$ |

### C.1. General Terms

Table 5 describes the correspondence between the key concepts from the courtroom analogy and the corresponding components in MoDEX. Below, we provide a detailed explanation of each correspondence.

- **Case:** In the courtroom, the case refers to the instance under evaluation by the court. In MoDEX, this corresponds to the input sample $\mathbf{x}_i$, which is the data instance processed by the model for evaluation.

- **Advocate:** In the courtroom, each advocate represents an individual who argues for a specific position or class. In MoDEX, this role is played by a Dirichlet expert with latent index $L_i \in \{1, \ldots, K\}$, which generates a probabilistic opinion over the class probabilities for the given input.

- **Advocate's Opinion:** An advocate's opinion in the courtroom reflects their interpretation of the case. In MoDEX, the opinion of the Dirichlet expert with latent index $L_i = k$ is represented by a Dirichlet distribution parameterized by

$\boldsymbol{\alpha}_k(\mathbf{x}_i)$. This parameter combines shared evidence with class-specific advocacy, modeling the expert's belief over the latent class probability vector.

- **Shared Evidence:** In the courtroom, shared evidence refers to the objective facts that are available to, and must be acknowledged by, all advocates. Such evidence is neutral and independent of any particular stance, including witness testimonies, physical evidence, or official documents that both the defense and the prosecution must consider when arguing the case. In MoDEX, this notion is captured by the shared base evidence parameter $\boldsymbol{\alpha}(\mathbf{x}_i)$, which represents information that is common across all Dirichlet experts. This parameter encodes the objective features of the input that are agreed upon by all advocates, serving as a common evidential foundation upon which class-specific arguments are built.

- **Class-Specific Advocacy:** While advocates rely on shared evidence, each advocate typically emphasizes certain aspects of the evidence or interprets them in a way that supports their own position. For instance, in the courtroom, the defense may stress alibi-related evidence to argue that the defendant was not present at the crime scene, whereas the prosecution may highlight evidence indicating the defendant's presence or motive. In MoDEX, this behavior is modeled through the class-specific advocacy parameter $\tau_k(\mathbf{x}_i)$. These parameters allow each Dirichlet expert to place additional emphasis on its own class beyond the shared evidence, thereby shaping class-specific beliefs over the latent class-probability vector.

- **Advocacy Plausibility:** In the courtroom, not all advocates' arguments are treated equally persuasively; instead, each opinion is evaluated in terms of its credibility or plausibility given the case at hand. Some arguments may be deemed more convincing based on their coherence with the evidence or their relevance to the case. This aspect is reflected in MoDEX by the advocacy plausibility weights $\boldsymbol{\omega}(\mathbf{x}_i)$, which quantify the relative credibility assigned to each advocate's opinion for a given input. These weights determine how strongly each class-specific belief contributes to the aggregated predictive distribution, playing a central role in the deliberation process that leads to the final verdict.

## C.2. Prediction and Uncertainty Measures

Table 6 summarizes how prediction and uncertainty measures in MoDEX correspond to concepts in the courtroom analogy. The correspondences are described below.

- **Belief Distribution Before Deliberation:** Before deliberation, each advocate holds an opinion about the outcome of the case, based on their interpretation of the evidence. In MoDEX, this is represented by the second-order distribution $\hat{p}(\boldsymbol{\pi}^\star \mid \mathbf{x}^\star)$ which captures the uncertainty across the different advocates' opinions before they are aggregated.

- **Final Verdict Distribution:** After deliberation, all advocates' opinions are aggregated into a single final decision. In MoDEX, this is represented by $\hat{p}(y^\star \mid \mathbf{x}^\star)$, which is a consolidated predictive distribution that encodes the court's overall belief over all possible verdicts, including residual uncertainty.

- **Final Verdict:** The final verdict corresponds to the deterministic decision issued by the court after deliberation. In MoDEX, this is given by the predicted label $\hat{y}(\mathbf{x}^\star)$, obtained by selecting the most probable class under the final verdict distribution.

- **Residual Verdict Ambiguity:** After deliberation, some uncertainty may still remain about the outcome. This residual uncertainty is captured by aleatoric uncertainty, $\mathrm{AU}(\mathbf{x}^\star)$, in MoDEX, which reflects the irreducible ambiguity in the final decision, based on the inherent variability in the data or the case itself.

- **Pre-deliberation Belief Uncertainty:** Before deliberation, there is uncertainty in the collection of advocate beliefs. In MoDEX, this is represented by epistemic uncertainty, $\mathrm{EU}(\mathbf{x}^\star)$, which accounts for both the internal imprecision within each advocate's belief and the disagreement between the different advocates.

- **Inter-advocate Disagreement:** This uncertainty arises when different advocates hold conflicting but confident opinions. In MoDEX, this is captured by $\mathrm{EU}_{\mathrm{inter}}(\mathbf{x}^\star)$, which quantifies disagreement across advocates.

- **Intra-advocate Imprecision:** Even within a single advocate's belief, uncertainty may arise due to weak or ambiguous evidence. In MoDEX, this is captured by $\mathrm{EU}_{\mathrm{intra}}(\mathbf{x}^\star)$, which reflects imprecision within individual advocates.

## C.3. Why the Courtroom Analogy is Useful

**Interpretability and extensibility.** The courtroom analogy provides more than an intuitive narrative; it offers a structured and operational framework for interpreting and extending single-pass uncertainty-aware classification models. By explicitly separating shared evidence, class-specific advocacy, and advocacy plausibility, MoDEX disentangles different sources of uncertainty in a manner that is both semantically meaningful and mathematically grounded. This structure allows practitioners to diagnose whether uncertainty arises from insufficient evidence, conflicting expert opinions, or imprecision within individual experts.

Beyond interpretability, the decomposition also facilitates extensibility. Because each courtroom concept corresponds to a well-defined modeling component, the framework can be naturally adapted to new settings by modifying specific elements without redesigning the entire model. For example, uncertainty-aware domain adaptation can be interpreted as reweighting advocacy plausibility based on domain relevance, while multi-view learning can be viewed as conducting parallel courtroom trials under different evidence sources and aggregating their outcomes. As a result, the courtroom analogy serves as a general design principle for constructing and extending uncertainty-aware models, rather than a task-specific analogy tied solely to MoDEX.

**Practical implications of epistemic uncertainty decomposition.** The decomposition of epistemic uncertainty into inter-expert and intra-expert components, formalized in Proposition 5.4, provides practical diagnostic value. High inter-expert uncertainty indicates disagreement among plausible class hypotheses. In real-world applications, this may suggest that the decision boundary or class semantics are ambiguous, and the appropriate response may be to refine class definitions, improve annotation guidelines, collect additional annotations, or defer the decision to human experts. In contrast, high intra-expert uncertainty reflects imprecision within individual expert opinions, often caused by weak, insufficient, or degraded input evidence. This type of uncertainty may arise when the input is corrupted, shifted from the training distribution, or simply difficult for the model to interpret reliably. In such cases, suitable interventions include improving input quality, applying better preprocessing, acquiring cleaner observations, or treating the sample as potentially affected by a distribution shift.

Thus, the inter- and intra-expert decomposition is not merely a mathematical reformulation of epistemic uncertainty. It provides practical value as a structured diagnostic tool for understanding why the model is uncertain and for selecting appropriate downstream actions based on the source of uncertainty.

# D. Proofs of the Theorems

In this section, we provide detailed proofs for Theorem 5.1, Proposition 5.2, Theorem 5.3, and Proposition 5.4.

## D.1. Proof of Theorem 5.1

*Proof.* The class probability distribution induced by MoDEX for test input $\mathbf{x}^\star$ is given by:

$$\hat{p}(\boldsymbol{\pi}^\star \mid \mathbf{x}^\star) = \sum_{k=1}^{K} \omega_k(\mathbf{x}^\star) \operatorname{Dir}(\boldsymbol{\pi}^\star \mid \boldsymbol{\alpha}(\mathbf{x}^\star) + \tau_k(\mathbf{x}^\star)\mathbf{e}_k),$$

where $\mathbf{e}_k$ denotes the $k$-th standard basis vector in $\mathbb{R}^K$.

**Reduction to $\mathcal{F}$-EDL.** Assume $\tau_k(\mathbf{x}^\star) = \tau(\mathbf{x}^\star)$ for all $k \in [K]$. Under this assumption, the predictive distribution for $\mathbf{x}^\star$ reduces to

$$\hat{p}(\boldsymbol{\pi} \mid \mathbf{x}^\star) = \sum_{k=1}^{K} \omega_k(\mathbf{x}^\star)\operatorname{Dir}(\boldsymbol{\pi}^\star \mid \boldsymbol{\alpha}(\mathbf{x}^\star) + \tau(\mathbf{x}^\star)\mathbf{e}_k).$$

This expression coincides with the mixture-form predictive distribution induced by $\mathcal{F}$-EDL for $\mathbf{x}^\star$, as characterized in Theorem 4.4 of Yoon & Kim (2026).

Therefore, when $\tau_k(\mathbf{x}^\star) = \tau(\mathbf{x}^\star)$ for all $k \in [K]$, the predictive distribution of MoDEX reduces to that of $\mathcal{F}$-EDL, that is,

$$\hat{p}(\boldsymbol{\pi}^\star \mid \mathbf{x}^\star) = \operatorname{FD}(\boldsymbol{\pi}^\star \mid \boldsymbol{\alpha}(\mathbf{x}^\star), \boldsymbol{\omega}(\mathbf{x}^\star), \tau(\mathbf{x}^\star)) = \hat{p}_{\mathcal{F}\text{-EDL}}(\boldsymbol{\pi}^\star \mid \mathbf{x}^\star),$$

where $\mathrm{FD}(\cdot \mid \boldsymbol{\alpha}(\mathbf{x}^\star), \boldsymbol{\omega}(\mathbf{x}^\star), \tau(\mathbf{x}^\star))$ denotes the probability density function of the flexible Dirichlet (Ongaro & Migliorati, 2013) distribution.

**Further reduction to EDL.** Assume additionally that $\tau(\mathbf{x}^\star) = 1$ and $\boldsymbol{\omega}(\mathbf{x}^\star) = \boldsymbol{\alpha}(\mathbf{x}^\star)/\|\boldsymbol{\alpha}(\mathbf{x}^\star)\|_1$. By Theorem 4.3 of the Yoon & Kim (2026) and its proof, substituting these conditions into the $\mathcal{F}$-EDL predictive distribution yields:

$$\hat{p}(\boldsymbol{\pi}^\star \mid \mathbf{x}^\star) = \mathrm{Dir}(\boldsymbol{\pi}^\star \mid \boldsymbol{\alpha}(\mathbf{x}^\star)) = \hat{p}_{\mathrm{EDL}}(\boldsymbol{\pi}^\star \mid \mathbf{x}^\star).$$

This completes the proof. $\square$

### D.2. Proof of Proposition 5.2

*Proof.* We show that the predictive distribution induced by MoDEX for a test input $\mathbf{x}^\star$ can be expressed as a weighted ensemble of EDL experts.

By definition, the MoDEX predictive distribution is given by:

$$\hat{p}(y^\star = k \mid \mathbf{x}^\star) = \int p(y^\star = k \mid \boldsymbol{\pi}^\star)\hat{p}(\boldsymbol{\pi}^\star \mid \mathbf{x}^\star)d\boldsymbol{\pi}^\star, \quad \forall k \in [K].$$

Since $p(y^\star = k \mid \boldsymbol{\pi}^\star) = \pi_k$, it suffices to compute the expectation of $\pi_k^\star$ under the MoDEX induced distribution.

By construction, MoDEX defines the following mixture of Dirichlet distributions:

$$\hat{p}(\boldsymbol{\pi}^\star \mid \mathbf{x}^\star) = \sum_{j=1}^{K} \omega_j(\mathbf{x}^\star)\mathrm{Dir}(\boldsymbol{\pi}^\star \mid \boldsymbol{\alpha}(\mathbf{x}^\star) + \tau_j(\mathbf{x}^\star)\mathbf{e}_j).$$

For any class $k \in [K]$, we have:

$$\begin{aligned}
\hat{p}(y^\star = k \mid \mathbf{x}^\star) &= \int \pi_k^\star \,\hat{p}(\boldsymbol{\pi}^\star \mid \mathbf{x}^\star)d\boldsymbol{\pi}^\star \\
&= \int \pi_k^\star \left( \sum_{j=1}^{K} \omega_j(\mathbf{x}^\star)\mathrm{Dir}\big(\boldsymbol{\pi}^\star \mid \boldsymbol{\alpha}(\mathbf{x}^\star) + \tau_j(\mathbf{x}^\star)\mathbf{e}_j\big) \right) d\boldsymbol{\pi}^\star \\
&= \sum_{j=1}^{K} \omega_j(\mathbf{x}^\star) \int \pi_k^\star \times \mathrm{Dir}\big(\boldsymbol{\pi}^\star \mid \boldsymbol{\alpha}(\mathbf{x}^\star) + \tau_j(\mathbf{x}^\star)\mathbf{e}_j\big)d\boldsymbol{\pi}^\star \\
&= \sum_{j=1}^{K} \omega_j(\mathbf{x}^\star) \times \mathbb{E}_{\boldsymbol{\pi}^\star \sim \mathrm{Dir}(\boldsymbol{\alpha}(\mathbf{x}^\star)+\tau_j(\mathbf{x}^\star)\mathbf{e}_j)}[\pi_k^\star].
\end{aligned}$$

Now, define the predictive distribution induced by the $j$-th EDL expert for $\mathbf{x}^\star$ as:

$$\begin{aligned}
\hat{p}_{\mathrm{EDL}}^{(j)}(y^\star = k \mid \mathbf{x}^\star) &:= \int p(y^\star = k \mid \boldsymbol{\pi}^\star)\mathrm{Dir}(\boldsymbol{\pi}^\star \mid \boldsymbol{\alpha}(\mathbf{x}^\star) + \tau_j(\mathbf{x}^\star)\mathbf{e}_j)d\boldsymbol{\pi}^\star \\
&= \mathbb{E}_{\boldsymbol{\pi}^\star \sim \mathrm{Dir}(\boldsymbol{\alpha}(\mathbf{x}^\star)+\tau_j(\mathbf{x}^\star)\mathbf{e}_j)}[\pi_k^\star].
\end{aligned}$$

Substituting this definition into the previous expression, we obtain:

$$\hat{p}(y^\star = k \mid \mathbf{x}^\star) = \sum_{j=1}^{K} \omega_j(\mathbf{x}^\star) \times \hat{p}_{\mathrm{EDL}}^{(j)}(y^\star = k \mid \mathbf{x}^\star).$$

Stacking the class probabilities, we obtain the final predictive distribution:

$$\hat{p}(y^\star \mid \mathbf{x}^\star) = \sum_{j=1}^{K} \omega_j(\mathbf{x}^\star) \times \hat{p}_{\mathrm{EDL}}^{(j)}(y^\star \mid \mathbf{x}^\star).$$

Thus, we have shown that the MoDEX predictive distribution is the weighted ensemble of the EDL experts, with the weights given by the mixture coefficients $\omega_j(\mathbf{x}^\star)$ and the individual expert predictions $\hat{p}_{\mathrm{EDL}}^{(j)}(y^\star \mid \mathbf{x}^\star)$. $\square$

### D.3. Proof of Theorem 5.3

*Proof.* We show that the predictive distribution induced by MoDEX for a test input $\mathbf{x}^\star$ can be expressed as an input-adaptive mixture of an EDL predictor with base evidence and a softmax-based predictor.

By the equivalence between the structured mixture of Dirichlet distributions in MoDEX and the extended Flexible Dirichlet (EFD) distribution, the predictive distribution for $\mathbf{x}^\star$ can be written as:

$$\hat{p}(y^\star = k \mid \mathbf{x}^\star) = \mathbb{E}_{\boldsymbol{\pi}^\star \sim \text{EFD}(\boldsymbol{\alpha}(\mathbf{x}^\star), \boldsymbol{\omega}(\mathbf{x}^\star), \boldsymbol{\tau}(\mathbf{x}^\star))}[\pi_k^\star]$$

Using the closed-form expression for the first-order moments of the EFD distribution (see Appendix A), we obtain, for all $k \in [K]$,

$$\hat{p}(y^\star = k \mid \mathbf{x}^\star) = \alpha_k(\mathbf{x}^\star)\kappa_1(\mathbf{x}^\star) + \frac{\tau_k(\mathbf{x}^\star)\,\omega_k(\mathbf{x}^\star)}{\|\boldsymbol{\alpha}(\mathbf{x}^\star)\|_1 + \tau_k(\mathbf{x}^\star)},$$

where $\alpha_k(\mathbf{x}^\star)$ denotes the $k$-th component of $\boldsymbol{\alpha}(\mathbf{x}^\star)$, and $\kappa_1(\mathbf{x}^\star)$ is defined as:

$$\kappa_1(\mathbf{x}^\star) = \sum_{j=1}^{K} \frac{\omega_j(\mathbf{x}^\star)}{\|\boldsymbol{\alpha}(\mathbf{x}^\star)\|_1 + \tau_j(\mathbf{x}^\star)}$$

Now, define the predictive distribution of the EDL predictor with base evidence as:

$$\hat{p}_{\text{EDL}}(y^\star \mid \mathbf{x}^\star) := \text{Cat}\left(\frac{\boldsymbol{\alpha}(\mathbf{x}^\star)}{\|\boldsymbol{\alpha}(\mathbf{x}^\star)\|_1}\right), \quad \text{i.e.,} \quad \hat{p}_{\text{EDL}}(y^\star = k \mid \mathbf{x}^\star) := \frac{\alpha_k(\mathbf{x}^\star)}{\|\boldsymbol{\alpha}(\mathbf{x}^\star)\|_1}, \forall k \in [K],$$

and define the predictive distribution of the softmax-based predictors as:

$$\hat{p}_{\text{SM}}(y^\star \mid \mathbf{x}^\star) := \text{Cat}(\boldsymbol{\omega}(\mathbf{x}^\star)), \quad \text{i.e.,} \quad \hat{p}_{\text{SM}}(y^\star = k \mid \mathbf{x}^\star) := \omega_k(\mathbf{x}^\star), \forall k \in [K].$$

Plugging these into the predictive distribution of MoDEX yields

$$\hat{p}(y^\star = k \mid \mathbf{x}^\star) = \left(\sum_{j=1}^{K} \frac{\|\boldsymbol{\alpha}(\mathbf{x}^\star)\|_1 \omega_j(\mathbf{x}^\star)}{\|\boldsymbol{\alpha}(\mathbf{x}^\star)\|_1 + \tau_j(\mathbf{x}^\star)}\right) \times \hat{p}_{\text{EDL}}(y^\star = k \mid \mathbf{x}^\star) + \left(\frac{\tau_k(\mathbf{x}^\star)}{\|\boldsymbol{\alpha}(\mathbf{x}^\star)\|_1 + \tau_k(\mathbf{x}^\star)}\right) \times \hat{p}_{\text{SM}}(y^\star = k \mid \mathbf{x}^\star),$$

Stacking the class probabilities, we obtain the final predictive distribution:

$$\hat{p}(y^\star \mid \mathbf{x}^\star) = \lambda_{\text{EDL}}(\mathbf{x}^\star) \times \hat{p}_{\text{EDL}}(y^\star \mid \mathbf{x}^\star) + \boldsymbol{\lambda}_{\text{SM}}(\mathbf{x}^\star) \odot \hat{p}_{\text{SM}}(y^\star \mid \mathbf{x}^\star),$$

where $\odot$ denotes the Hadamard (elementwise) product, and the mixture weights are:

$$\lambda_{\text{EDL}}(\mathbf{x}^\star) := \sum_{j=1}^{K} \frac{\|\boldsymbol{\alpha}(\mathbf{x}^\star)\|_1 \,\omega_j(\mathbf{x}^\star)}{\|\boldsymbol{\alpha}(\mathbf{x}^\star)\|_1 + \tau_j(\mathbf{x}^\star)}, \quad \boldsymbol{\lambda}_{\text{SM}}(\mathbf{x}^\star) := \left[\frac{\tau_k(\mathbf{x}^\star)}{\|\boldsymbol{\alpha}(\mathbf{x}^\star)\|_1 + \tau_k(\mathbf{x}^\star)}\right]_{k=1}^{K}.$$

Thus, we have shown that the MoDEX predictive distribution is the class- and input-dependent mixture of the EDL predictor with base-evidence, $\hat{p}_{\text{EDL}}(y^\star \mid \mathbf{x}^\star)$, and the softmax predictor $\hat{p}_{\text{SM}}(y^\star \mid \mathbf{x}^\star)$. $\qquad\square$

### D.4. Proof of Proposition 5.4

*Proof.* Let $L^\star \in \{1, \dots, K\}$ denote the latent expert index for the test input $\mathbf{x}^\star$ with $\mathbb{P}(L^\star = k \mid \mathbf{x}^\star) = \omega_k(\mathbf{x}^\star), \forall k \in [K]$.

Conditioned on $L^\star = k$, MoDEX induces a Dirichlet distribution over the latent class probability vector $\boldsymbol{\pi}^\star$,

$$\boldsymbol{\pi}^\star \mid (L^\star = k, \mathbf{x}^\star) \sim \text{Dir}\left(\boldsymbol{\alpha}_k(\mathbf{x}^\star)\right),$$

where $\boldsymbol{\alpha}_k(\mathbf{x}^\star) := \boldsymbol{\alpha}(\mathbf{x}^\star) + \tau_k(\mathbf{x}^\star)\mathbf{e}_k$.

We define the epistemic uncertainty induced by MoDEX for the test input $\mathbf{x}^\star$ as the total variance (i.e., the trace of the covariance matrix) of the latent class probability vector $\boldsymbol{\pi}^\star$:

$$\mathrm{EU}(\mathbf{x}^\star) := \mathrm{tr}\left(\mathrm{Cov}_{\boldsymbol{\pi}^\star \sim \hat{p}(\boldsymbol{\pi}^\star | \mathbf{x}^\star)}[\boldsymbol{\pi}^\star]\right),$$

where $\hat{p}(\boldsymbol{\pi}^\star \mid \mathbf{x}^\star)$ is the class probability distribution for $\mathbf{x}^\star$ induced by MoDEX.

Applying the law of total covariance with respect to the latent expert index $L^\star$, we decompose the total uncertainty into the expected within-expert variability and the variability of expert-wise predictive means:

$$\mathrm{Cov}_{\boldsymbol{\pi}^\star \sim \hat{p}(\boldsymbol{\pi}^\star | \mathbf{x}^\star)}[\boldsymbol{\pi}^\star] = \mathbb{E}_{L^\star \sim \mathrm{Cat}(\boldsymbol{\omega}(\mathbf{x}^\star))}\left[\mathrm{Cov}_{\boldsymbol{\pi}^\star \sim \hat{p}(\boldsymbol{\pi}^\star | \mathbf{x}^\star)}[\boldsymbol{\pi}^\star] \mid L^\star\right] + \mathrm{Cov}_{L \sim \mathrm{Cat}(\boldsymbol{\omega}(\mathbf{x}^\star))}\left(\mathbb{E}_{\boldsymbol{\pi}^\star \sim \hat{p}(\boldsymbol{\pi}^\star | \mathbf{x}^\star)}[\boldsymbol{\pi}^\star \mid L^\star]\right).$$

The first term can be expressed as:

$$\mathbb{E}_{L^\star \sim \mathrm{Cat}(\boldsymbol{\omega}^\star)}\left[\mathrm{Cov}_{\boldsymbol{\pi}^\star \sim \hat{p}(\boldsymbol{\pi}^\star | \mathbf{x}^\star)}[\boldsymbol{\pi}^\star] \mid L^\star\right] = \sum_{k=1}^{K} P(L^\star = k \mid \mathbf{x}^\star) \times \mathrm{Cov}_{\boldsymbol{\pi}^\star \sim \hat{p}(\boldsymbol{\pi}^\star | L^\star = k, \mathbf{x}^\star)}[\boldsymbol{\pi}^\star]$$

$$= \sum_{k=1}^{K} \omega_k(\mathbf{x}^\star) \times \mathrm{Cov}_{\boldsymbol{\pi}^\star \sim \mathrm{Dir}(\boldsymbol{\alpha}_k(\mathbf{x}^\star))}[\boldsymbol{\pi}^\star].$$

This term captures the expected variability arising from insufficient evidence within each Dirichlet expert and therefore corresponds to the intra-expert uncertainty.

For the second term, define the mean class probability vector of the $k$-th expert as

$$\boldsymbol{\mu}^{(k)}(\mathbf{x}^\star) := \mathbb{E}_{\boldsymbol{\pi}^\star \sim \mathrm{Dir}(\boldsymbol{\alpha}_k(\mathbf{x}^\star))}[\boldsymbol{\pi}^\star],$$

and the aggregated mean vector as:

$$\bar{\boldsymbol{\mu}}(\mathbf{x}^\star) := \sum_{k=1}^{K} \omega_k(\mathbf{x}^\star)\boldsymbol{\mu}^{(k)}(\mathbf{x}^\star).$$

Then, the second term can be expressed as:

$$\mathrm{Cov}_{L^\star \sim \mathrm{Cat}(\boldsymbol{\omega}(\mathbf{x}^\star))}\left(\mathbb{E}_{\boldsymbol{\pi}^\star \sim \hat{p}(\boldsymbol{\pi}^\star | \mathbf{x}^\star)}[\boldsymbol{\pi}^\star \mid L^\star]\right) = \mathrm{Cov}_{L^\star \sim \mathrm{Cat}(\boldsymbol{\omega}(\mathbf{x}^\star))}(\boldsymbol{\mu}^{(L^\star)})$$

$$= \mathbb{E}_{L^\star \sim \mathrm{Cat}(\boldsymbol{\omega}(\mathbf{x}^\star))}\left[(\boldsymbol{\mu}^{(L^\star)}(\mathbf{x}^\star) - \mathbb{E}[\boldsymbol{\mu}^{(L^\star)}(\mathbf{x}^\star)])(\boldsymbol{\mu}^{(L^\star)}(\mathbf{x}^\star) - \mathbb{E}[\boldsymbol{\mu}^{(L^\star)}(\mathbf{x}^\star)])^T\right]$$

$$= \sum_{k=1}^{K} \omega_k(\mathbf{x}^\star) \times \left(\boldsymbol{\mu}^{(k)}(\mathbf{x}^\star) - \bar{\boldsymbol{\mu}}(\mathbf{x}^\star)\right)(\boldsymbol{\mu}^{(k)}(\mathbf{x}^\star) - \bar{\boldsymbol{\mu}}(\mathbf{x}^\star))^T,$$

which quantifies the disagreement among expert-wise predictive means and corresponds to the inter-expert uncertainty.

Combining the two terms yields,

$$\mathrm{Cov}_{\boldsymbol{\pi}^\star \sim \hat{p}(\boldsymbol{\pi}^\star | \mathbf{x}^\star)}[\boldsymbol{\pi}^\star] = \sum_{k=1}^{K} \omega_k(\mathbf{x}^\star) \times \mathrm{Cov}_{\boldsymbol{\pi}^\star \sim \mathrm{Dir}(\boldsymbol{\alpha}_k(\mathbf{x}^\star))}[\boldsymbol{\pi}^\star] + \sum_{k=1}^{K} \omega_k(\mathbf{x}^\star) \times \left((\boldsymbol{\mu}^{(k)}(\mathbf{x}^\star) - \bar{\boldsymbol{\mu}}(\mathbf{x}^\star))(\boldsymbol{\mu}^{(k)}(\mathbf{x}^\star) - \bar{\boldsymbol{\mu}}(\mathbf{x}^\star))^T\right)$$

Taking the trace on both sides and using the linearity of the trace, we obtain

$$\mathrm{EU}(\mathbf{x}^\star) = \sum_{k=1}^{K} \omega_k(\mathbf{x}^\star)\mathrm{tr}\left(\mathrm{Cov}_{\boldsymbol{\pi}^\star \sim \mathrm{Dir}(\boldsymbol{\alpha}_k(\mathbf{x}^\star))}[\boldsymbol{\pi}^\star]\right) + \sum_{k=1}^{K} \omega_k(\mathbf{x}^\star)\mathrm{tr}\left((\boldsymbol{\mu}^{(k)}(\mathbf{x}^\star) - \bar{\boldsymbol{\mu}}(\mathbf{x}^\star))(\boldsymbol{\mu}^{(k)}(\mathbf{x}^\star) - \bar{\boldsymbol{\mu}}(\mathbf{x}^\star))^T\right).$$

We now further elaborate on each component explicitly.

First, the inter-expert uncertainty can be written as:

$$\mathrm{EU}_{\mathrm{inter}}(\mathbf{x}^\star) := \sum_{k=1}^{K} \omega_k(\mathbf{x}^\star) \,\mathrm{tr}\Big((\boldsymbol{\mu}^{(k)}(\mathbf{x}^\star) - \bar{\boldsymbol{\mu}}(\mathbf{x}^\star))(\boldsymbol{\mu}^{(k)}(\mathbf{x}^\star) - \bar{\boldsymbol{\mu}}(\mathbf{x}^\star))^T\Big) = \sum_{k=1}^{K} \omega_k(\mathbf{x}^\star)\big\|\boldsymbol{\mu}^{(k)}(\mathbf{x}^\star) - \bar{\boldsymbol{\mu}}(\mathbf{x}^\star)\big\|_2^2,$$

where we used the identity $\mathrm{tr}(\mathbf{v}\mathbf{v}^T) = \mathrm{tr}(\mathbf{v}^T\mathbf{v}) = \mathbf{v}^T\mathbf{v} = \|\mathbf{v}\|_2^2$ for any vector $\mathbf{v}$.

Using the standard weighted variance identity, the inter-expert uncertainty can also be written in an equivalent form that highlights the pairwise disagreement between the experts:

$$\mathrm{EU}_{\mathrm{inter}}(\mathbf{x}^\star) = \frac{1}{2}\sum_{k=1}^{K}\sum_{j=1}^{K} \omega_k(\mathbf{x}^\star)\omega_j(\mathbf{x}^\star)\|\boldsymbol{\mu}^{(k)} - \boldsymbol{\mu}^{(j)}\|_2^2.$$

Second, the intra-expert uncertainty can be written as:

$$\mathrm{EU}_{\mathrm{intra}}(\mathbf{x}^\star) := \sum_{k=1}^{K} \omega_k(\mathbf{x}^\star)\mathrm{tr}\Big(\mathrm{Cov}_{\boldsymbol{\pi}^\star \sim \mathrm{Dir}(\boldsymbol{\alpha}_k(\mathbf{x}^\star))}[\boldsymbol{\pi}^\star]\Big) = \sum_{k=1}^{K} \omega_k(\mathbf{x}^\star)\sum_{j=1}^{K} \mathrm{Var}_{\boldsymbol{\pi}^\star \sim \mathrm{Dir}(\boldsymbol{\alpha}_k(\mathbf{x}^\star))}[\pi_j^\star],$$

where the last equality follows the fact that the trace of the covariance matrix equals the sum of its diagonal entries.

This completes the proof. □

# E. Additional Explanations of the Experiments

In this section, we provide additional details on our experimental setup. First, we describe the datasets used in our experiments (Appendix E.1). Second, we present additional implementation details (Appendix E.2).

### E.1. Datasets

We describe the datasets used in our experiments, all of which are publicly available benchmarks.

**CIFAR-10** (Krizhevsky et al., 2009) is a standard benchmark for image classification and UQ. It consists of color images from 10 classes covering common animals and objects: airplane, automobile, bird, cat, deer, dog, frog, horse, ship, and truck. The dataset comprises 50,000 training images and 10,000 test images, with each image represented as $3 \times 32 \times 32$ RGB tensor. CIFAR-10 is class-balanced, with an equal number of examples per class in both the training and test splits. In our experiments, the training set is split into training and validation subsets using a 0.95:0.05 ratio. CIFAR-10 serves as the primary ID dataset and is also used to construct CIFAR-10-LT, a long-tailed variant employed in our evaluations.

**CIFAR-100** (Krizhevsky et al., 2009) is a more fine-grained and challenging extension of the CIFAR-10 dataset, consisting of 100 classes that span diverse animal and object categories. The dataset contains 50,000 training images and 10,000 test images, with each image represented as a $3 \times 32 \times 32$ RGB tensor. In our experiments, the training set is split into training and validation subsets with the same 0.95:0.05 ratio. CIFAR-100 is used as the ID dataset for the standard setting, and its test set is additionally employed as an OOD dataset when CIFAR-10 or CIFAR-10-LT serves as the ID dataset.

**Street View House Number (SVHN)** (Netzer et al., 2011) is a real-world dataset consisting of cropped images of house numbers collected from Google Street View. The dataset includes 73,257 training images, 26,032 test images, and an additional set of 531,131 images, with each image represented as a $3 \times 32 \times 32$ RGB tensor. In our experiments, the test set of SVHN is used as the OOD dataset when CIFAR-10, CIFAR-10-LT, or CIFAR-100 serves as the ID dataset.

**Tiny-ImageNet-200 (TIN-200)** (Le & Yang, 2015) is a downsampled subset of the ImageNet dataset (Deng et al., 2009), consisting of 200 classes spanning diverse object categories. The dataset contains 100,000 training images, 10,000 validation images, and 10,000 test images, with each image represented as a $3 \times 64 \times 64$ RGB tensor. In our experiments, the test set of TIN-200 is used as the OOD dataset when CIFAR-100 serves as the ID dataset. To ensure compatibility with CIFAR-100, all TIN-200 images are resized to $3 \times 32 \times 32$ during pre-processing.

**CIFAR-10-LT** (Cui et al., 2019) is a long-tailed variant of the CIFAR-10 dataset designed to evaluate classification robustness under varying degrees of class imbalance. The severity level of imbalance is controlled by the imbalance factor $\rho$,

defined as the ratio between the number of samples in the most frequent (head) class and the least frequent (tail) class. In our experiments, we consider CIFAR-10-LT with two imbalance regimes: (i) mild imbalance with $\rho = 0.1$ and (ii) a severe imbalance with $\rho = 0.01$. Both settings are used as ID datasets to evaluate robustness under long-tailed conditions.

**CIFAR-10-C** (Hendrycks & Dietterich, 2019) is a corrupted variant of the CIFAR-10 dataset designed to evaluate model robustness under common distribution shifts induced by corruption and noise. It applies 19 types of corruptions—including Gaussian noise, shot noise, impulse noise, defocus blur, glass blur, motion blur, zoom blur, snow, frost, fog, brightness, contrast, elastic transform, pixelation, JPEG compression, speckle noise, Gaussian blur, splatter, and saturation—to the CIFAR-10 test images. Each corruption type is applied at five severity levels, $\mathcal{C} = \{1, 2, 3, 4, 5\}$, resulting in a total of 95 corrupted test sets (19 corruption types × 5 severity levels), each containing 10,000 images. In our experiments, CIFAR-10-C is used to evaluate distribution shift detection performance. Specifically, models are trained on CIFAR-10 as the ID dataset and evaluated on each corrupted test set, yielding 95 OOD evaluations. We report results averaged across corruption types for each severity level.

**Dirty-MNIST (DMNIST)** (Mukhoti et al., 2023) is a noisy variant of the MNIST (LeCun, 1998) dataset that contains artificially generated ambiguous digit images and is widely used to evaluate the robustness of UQ models under noisy ID settings. The dataset is constructed by combining the original MNIST dataset with Ambiguous-MNIST (AMNIST), which consists of synthetically generated digit images exhibiting varying levels of entropy. As a result, DMNIST contains 120,000 training images, comprising 60,000 MNIST samples and 60,000 AMNIST samples. In our experiments, we use the DMNIST dataset to create a long-tailed variant, referred to as DMNIST-LT, in order to simulate more challenging scenarios that involve both ambiguity and class imbalance.

**Dirty-MNIST-LT (DMNIST-LT)** is a long-tailed variant of DMNIST, introduced in our work, in which class distributions are artificially imbalanced. The degree of imbalance is controlled by an imbalance factor of $\rho = 0.01$ to evaluate the robustness of the proposed model under simultaneous noise and long-tailed conditions. This setting reflects a realistic industrial scenario, in which data are often both noisy—due to sensor imperfections—and highly imbalanced.

**Fashion-MNIST (FMNIST)** (Xiao et al., 2017) is a modern alternative to MNIST, consisting of grayscale images from 10 classes of fashion items, including clothing, footwear, and accessories. The dataset contains 60,000 training images and 10,000 test images, with each image having a resolution of $1 \times 28 \times 28$. In our experiments, FMNIST is used as the OOD dataset when DMNIST-LT serves as the ID dataset.

**CIFAR-10H** (Peterson et al., 2019) is a human-annotated extension of the CIFAR-10 test set that provides soft labels reflecting human perceptual uncertainty. For each test image, a label was collected from multiple human annotators, and the resulting empirical label distribution captures inherent ambiguity arising from visually similar or confusing classes. In our qualitative experiments, we use the CIFAR-10H human label distributions as a proxy for semantic ambiguity and leverage them to assess whether models can capture such ambiguity in an interpretable manner.

### E.2. Implementation Details

For fair comparison, we adhere to the experimental protocols established in recent EDL studies (Charpentier et al., 2020; Deng et al., 2023; Chen et al., 2024; Yoon & Kim, 2024; Chen et al., 2025; Yoon & Kim, 2026), particularly regarding the backbone architectures. Below, we detail the backbone architectures used for each ID dataset, as well as the design of the lightweight MLP heads. In our framework, the backbone network consists of the feature extractor $f_\psi$ and the concentration head $g_{\phi_1}$, which are shared with standard EDL baselines. The additional components—specifically the gating head $g_{\phi_2}$ and the advocacy-strength head $g_{\phi_3}$—are implemented as lightweight MLPs on top of the backbone.

**Backbones.** Consistent with established protocol in recent EDL literature, we use VGG-16 (Simonyan & Zisserman, 2014) as the backbone architecture when CIFAR-10 or CIFAR-10-LT serves as the ID dataset. For CIFAR-100, we adopt ResNet-18 (He et al., 2016) as the backbone. For DMNIST-LT, we employ a straightforward convolutional neural network (ConvNet) consisting of three convolutional layers followed by three fully connected layers.

**MLP Heads.** We implement the MLP heads as shallow, fully connected networks. Specifically, each head consists of a single fully connected layer for DMNIST and two fully connected layers for all other datasets.

**Optimization and Training.** We applied the Adam optimizer (Kingma & Ba, 2014) in conjunction with a StepLR scheduler for all experiments. A fixed batch size of $B = 64$ is used across all settings. Training is performed for up to 50 epochs on DMNIST and DMNIST-LT, and up to 200 epochs on the remaining datasets. Early stopping is applied based on the validation loss. Hyperparameters for both the proposed method and all baselines are selected via grid search. Detailed

*Table 7.* Implementation details and hyperparameter settings for all experiments. $T_{\max}$ denotes the maximum number of training epochs, $\eta$ is the learning rate, and Step Size refers to the step size in the StepLR scheduler. $L$ and $H$ represent the number of layers and the hidden dimension of the additional MLP heads, respectively. $\epsilon$ is the label smoothing parameter.

| ID Dataset (s) | Architecture | $T_{\max}$ | $\eta$ | Step Size | $L$ | $H$ | $\epsilon$ |
|---|---|---|---|---|---|---|---|
| DMNIST-LT | ConvNet | 50 | $10^{-3}$ | 20 | 1 | 3 | 0.1 |
| CIFAR-10 | VGG-16 | 200 | $5 \times 10^{-4}$ | 100 | 2 | 128 | 0.1 |
| CIFAR-10-LT | VGG-16 | 200 | $10^{-3}$ | 100 | 2 | 128 | 0.1 |
| CIFAR-100 | ResNet-18 | 200 | $10^{-4}$ | 100 | 2 | 128 | 0.1 |

hyperparameter configuration is outlined in Table 7.

### E.3. Computational Costs and Scalability

**Parameter Overhead.** We first quantify the parameter overhead introduced by the MoDEX-specific heads described in the implementation details. Specifically, we compare the number of trainable parameters in the base model with the additional parameters introduced by the gating head and the advocacy-strength head.

*Table 8.* Parameter overhead introduced by the additional MLP heads in each experimental setting. For each dataset, we report the number of trainable parameters in the base model $(f_\psi + g_{\phi_1})$, the additional parameters introduced by the extra MLP heads $(g_{\phi_2}, g_{\phi_3})$, and the resulting total parameter count, along with the relative increase in parameters.

| ID Dataset(s) | Base Params | Added Params | Total Params | Increase (%) |
|---|---|---|---|---|
| DMNIST-LT | 252,490 | 3,554 | 256,044 | 1.41 |
| CIFAR-10, CIFAR-10-LT | 14,857,546 | 134,420 | 14,991,966 | 0.90 |
| CIFAR-100 | 11,046,308 | 157,640 | 11,203,948 | 1.43 |

As shown in Table 8, the additional heads incur only a modest increase in model complexity: 1.41% for DMNIST-LT, 0.90% for CIFAR-10 and CIFAR-10-LT, and 1.43% on CIFAR-100. This indicates that MoDEX introduces its additional courtroom parameters with only a small parameter overhead over the EDL-style base model. In typical image classification settings, this overhead remains minor because the shared backbone dominates the total model size.

**Inference Time.** Beyond parameter overhead, we further measure the end-to-end inference time, including both prediction and uncertainty-measure computation. The experiment is conducted on the CIFAR-10 dataset with VGG-16 using an RTX 4060 GPU and a batch size of 64. We compare the inference cost of MoDEX with those of EDL methods, including EDL (Sensoy et al., 2018) and $\mathcal{F}$-EDL (Yoon & Kim, 2026), as well as representative sampling-based approaches, including MC Dropout (Gal & Ghahramani, 2016), Bayesian neural networks (BNN) (Blundell et al., 2015), and Deep Ensembles (Lakshminarayanan et al., 2017). For sampling-based methods, we use 10 Monte Carlo samples for MC Dropout and BNN, and 5 independently trained models for Deep Ensembles.

*Table 9.* End-to-end inference time comparison on CIFAR-10 with VGG-16. The reported time includes both prediction and uncertainty-measure computation.

| Method | Inference Time (ms/batch) $\downarrow$ |
|---|---|
| MC Dropout | $59.92 \pm 1.5$ |
| BNN | $77.85 \pm 3.5$ |
| Deep Ensembles | $34.85 \pm 1.5$ |
| EDL | $12.53 \pm 2.4$ |
| $\mathcal{F}$-EDL | $13.50 \pm 0.1$ |
| MoDEX | $13.94 \pm 3.1$ |

As shown in Table 9, MoDEX has an inference time comparable to those of EDL and $\mathcal{F}$-EDL, while being substantially faster than sampling-based UQ methods. This is because MoDEX computes its predictive mean and uncertainty measures in

closed form from a single forward pass, without requiring Monte Carlo sampling or multiple independently trained models. These results support the practical efficiency of MoDEX as a single-pass UQ method.

**Scalability with Respect to the Number of Classes.** MoDEX uses the structured decomposition $\alpha_k(\mathbf{x}) = \alpha(\mathbf{x}) + \tau_k(\mathbf{x})\mathbf{e}_k$, and therefore predicts only $\alpha(\mathbf{x})$, $\omega(\mathbf{x})$, and $\tau(\mathbf{x})$ for each input $\mathbf{x}$. Its output parameterization thus scales linearly with the number of classes, i.e., $\mathcal{O}(K)$, rather than quadratically as in a naive mixture of $K$ full Dirichlet concentration vectors. Compared with EDL, MoDEX only introduces a constant-factor increase through two additional lightweight heads.

In practice, this overhead remains small even in higher-class settings. For instance, with WideResNet-28-10 on TinyImageNet-200, MoDEX adds only 216K parameters to a 36.5M-parameter backbone, corresponding to approximately $0.59\%$ additional parameters. This indicates that MoDEX remains lightweight as the number of classes increases, since the shared backbone typically dominates the total model size.

# F. Additional Results for Quantitative Studies on UQ-Related Downstream Tasks

In this section, we present additional quantitative results to complement the main experimental findings. For ease of reference, the contents of each table are summarized below.

- Table 10: Comprehensive results for UQ-related downstream tasks under the standard setting with CIFAR-10.

- Table 11: AUROC scores for OOD detection under the standard setting.

- Table 12: Comprehensive results for distribution shift detection from CIFAR-10 to CIFAR-10-C.

- Table 13: AUROC scores for distribution shift detection from CIFAR-10 to CIFAR-10-C.

- Table 14: UQ-related downstream tasks under CIFAR-10-LT with mild imbalance ($\rho = 0.1$).

- Table 15: AUROC scores for UQ-related downstream tasks under the long-tailed setting.

- Table 17: Ablation study results on MoDEX parameters using the CIFAR-10 dataset.

- Table 18: Ablation study results for the regularization terms in the objective function using the CIFAR-10 dataset.

- Table 16: Comparison with non-EDL single-pass UQ methods under the long-tailed and noisy setting using DMNIST-LT.

- Table 19: Comparison with Deep Ensembles using CIFAR-10 dataset.

The corresponding results, along with a brief explanation and interpretation, are provided below.

*Table 10.* Comprehensive results for UQ-related downstream tasks in the standard setting with CIFAR-10 as the ID dataset. We additionally report results from PostNet, NatPN, DUQ, and RED. Baseline results are taken from prior work (Chen et al., 2025; Yoon & Kim, 2026).

| Method | Test.Acc. | Miscl. AUPR. | SVHN / CIFAR-100 |
|---|---|---|---|
| Dropout | 90.16 ±0.2 | 98.86 ±0.6 | 78.40 ±3.9 / 85.39 ±0.6 |
| PostNet | 87.82 ±0.1 | 97.46 ±0.1 | 83.76 ±0.5 / 87.07 ±0.9 |
| NatPN | 87.73 ±0.1 | 97.53 ±0.1 | 83.56 ±0.4 / 86.98 ±0.8 |
| DUQ | 89.33 ±0.2 | 97.89 ±0.3 | 80.23 ±3.4 / 84.75 ±1.1 |
| EDL | 88.48 ±0.3 | 98.74 ±0.1 | 82.32 ±1.2 / 87.13 ±0.3 |
| $\mathcal{I}$-EDL | 89.20 ±0.3 | 98.72 ±0.1 | 82.96 ±2.2 / 84.84 ±0.6 |
| RED | 89.43 ±0.3 | 98.82 ±0.1 | 82.85 ±2.4 / 87.84 ±0.5 |
| R-EDL | 90.09 ±0.3 | 98.98 ±0.1 | 85.00 ±1.2 / 87.73 ±0.3 |
| DAEDL | 91.11 ±0.2 | 99.08 ±0.0 | 85.54 ±1.4 / 88.19 ±0.1 |
| Re-EDL | 90.09 ±0.3 | 98.81 ±0.1 | 89.94 ±1.4 / 88.31 ±0.2 |
| $\mathcal{F}$-EDL | 91.19 ±0.2 | 99.10 ±0.0 | 91.20 ±1.3 / 88.37 ±0.3 |
| **MoDEX** | **92.46** ±0.2 | **99.18** ±0.0 | **91.58** ±0.4 / **89.28** ±0.3 |

*Table 11.* AUROC scores for OOD detection using epistemic uncertainty estimates in the standard setting are reported for two scenarios: (i) CIFAR-10 as the ID dataset with SVHN and CIFAR-100 (C-100) as OOD datasets, and (ii) CIFAR-100 as the ID dataset with SVHN and TinyImageNet (TIN-200) as OOD datasets.

| Method | ID: CIFAR-10 OOD: SVHN / C-100 | ID: CIFAR-100 OOD: SVHN / TIN |
|---|---|---|
| EDL | 81.06 ±4.5 / 80.63 ±1.0 | 63.95 ±3.4 / 65.32 ±2.3 |
| $\mathcal{I}$-EDL | 86.79 ±1.3 / 82.15 ±0.5 | 77.85 ±1.5 / 73.34 ±0.3 |
| R-EDL | 87.47 ±1.2 / 85.26 ±0.4 | 77.06 ±2.2 / 71.10 ±1.0 |
| DAEDL | 89.24 ±1.0 / 86.04 ±0.1 | 81.07 ±3.0 / 75.04 ±1.3 |
| Re-EDL | 92.22 ±1.1 / 86.67 ±0.1 | 77.89 ±2.2 / 74.67 ±0.6 |
| $\mathcal{F}$-EDL | 93.74 ±1.5 / 86.37 ±0.3 | 81.59 ±1.8 / 79.24 ±0.2 |
| **MoDEX** | **93.97** ±0.8 / **87.45** ±0.4 | **84.42** ±1.1 / **80.22** ±0.1 |

*Table 12.* Comprehensive results for distribution shift detection using CIFAR-10 as the ID dataset. We report AUPR scores for detecting distribution shifts from CIFAR-10 to CIFAR-10-C based on epistemic uncertainty. $\mathcal{C} \in \{1, 2, 3, 4, 5\}$ denotes corruption severity levels in CIFAR-10-C, averaged across 19 corruption types.

| Method | $\mathcal{C} = 1$ | $\mathcal{C} = 2$ | $\mathcal{C} = 3$ | $\mathcal{C} = 4$ | $\mathcal{C} = 5$ |
|---|---|---|---|---|---|
| MSP | 56.39 $\pm 0.7$ | 61.88 $\pm 1.1$ | 65.86 $\pm 1.3$ | 69.91 $\pm 1.5$ | 75.01 $\pm 1.8$ |
| EDL | 55.56 $\pm 0.7$ | 59.81 $\pm 1.1$ | 63.38 $\pm 1.4$ | 67.55 $\pm 1.4$ | 73.12 $\pm 1.3$ |
| $\mathcal{I}$-EDL | 56.35 $\pm 0.5$ | 61.19 $\pm 0.8$ | 65.23 $\pm 1.2$ | 69.45 $\pm 1.6$ | 74.91 $\pm 1.9$ |
| R-EDL | 57.17 $\pm 0.5$ | 62.93 $\pm 0.9$ | 67.33 $\pm 1.2$ | 71.72 $\pm 1.2$ | 76.80 $\pm 1.2$ |
| DAEDL | 55.90 $\pm 0.8$ | 60.10 $\pm 1.3$ | 63.69 $\pm 1.4$ | 67.78 $\pm 1.3$ | 73.45 $\pm 1.5$ |
| Re-EDL | 56.93 $\pm 0.6$ | 62.55 $\pm 0.9$ | 66.84 $\pm 1.2$ | 71.15 $\pm 1.3$ | 76.05 $\pm 1.5$ |
| $\mathcal{F}$-EDL | 58.16 $\pm 0.5$ | 64.06 $\pm 0.6$ | 68.44 $\pm 0.8$ | 73.07 $\pm 0.9$ | 78.52 $\pm 1.1$ |
| **MoDEX** | **58.72** $\pm 0.5$ | **65.42** $\pm 0.3$ | **70.24** $\pm 0.5$ | **74.98** $\pm 0.5$ | **80.63** $\pm 0.5$ |

*Table 13.* AUROC scores for distribution shift detection from CIFAR-10 to CIFAR-10-C using epistemic uncertainty estimates.

| Method | $\mathcal{C} = 1$ | $\mathcal{C} = 2$ | $\mathcal{C} = 3$ | $\mathcal{C} = 4$ | $\mathcal{C} = 5$ |
|---|---|---|---|---|---|
| EDL | 55.99 $\pm 0.4$ | 60.47 $\pm 0.7$ | 64.13 $\pm 0.9$ | 68.21 $\pm 1.0$ | 73.41 $\pm 0.9$ |
| $\mathcal{I}$-EDL | 56.58 $\pm 0.4$ | 61.39 $\pm 0.7$ | 65.23 $\pm 1.1$ | 69.26 $\pm 1.5$ | 74.31 $\pm 1.8$ |
| R-EDL | 56.97 $\pm 0.6$ | 62.42 $\pm 1.0$ | 66.51 $\pm 1.2$ | 70.70 $\pm 1.2$ | 75.50 $\pm 1.2$ |
| DAEDL | 56.71 $\pm 0.2$ | 61.39 $\pm 0.4$ | 65.11 $\pm 0.4$ | 69.20 $\pm 0.4$ | 74.58 $\pm 0.6$ |
| Re-EDL | 56.60 $\pm 0.4$ | 61.85 $\pm 0.7$ | 65.84 $\pm 0.9$ | 69.96 $\pm 1.0$ | 74.57 $\pm 1.2$ |
| $\mathcal{F}$-EDL | 58.67 $\pm 0.3$ | 64.50 $\pm 0.4$ | 68.52 $\pm 0.5$ | 72.83 $\pm 0.6$ | 77.92 $\pm 0.8$ |
| **MoDEX** | **59.41** $\pm 0.2$ | **65.90** $\pm 0.3$ | **70.26** $\pm 0.4$ | **74.67** $\pm 0.5$ | **79.88** $\pm 0.5$ |

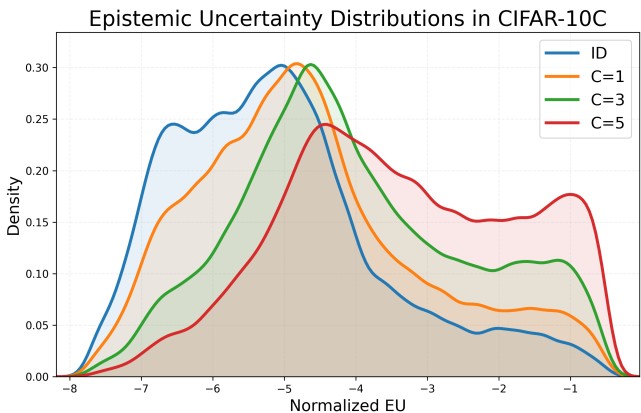

*Figure 2.* Qualitative visualization of epistemic uncertainty on CIFAR-10-C. MoDEX assigns higher epistemic uncertainty across corruption severities $\mathcal{C} = 1, 3, 5$, consistent with the expected behavior under increasingly severe distribution shifts.

**F.1. Standard Setting**

Table 10 presents comprehensive results for UQ-related downstream tasks in the standard setting, using CIFAR-10 as the ID dataset. In this table, we compare MoDEX with additional baselines, including DUQ (Van Amersfoort et al., 2020) and Posterior Network (PostNet) (Charpentier et al., 2020), Natural Posterior Network (NatPN) (Charpentier et al., 2021), and Regularized Evidential model (RED) (Pandey & Yu, 2023), alongside state-of-the-art EDL approaches. DUQ represents deterministic UQ methods and is categorized as an implicit second-order UQ method in our taxonomy. In contrast, PostNet and NatPN are explicit second-order UQ methods that enhance epistemic uncertainty estimation via feature-space density modeling. RED is an EDL-based model that employs theoretically motivated evidence activation functions together with a principled regularization scheme. Overall, the results show that MoDEX significantly outperforms all other models, highlighting its strong competitiveness across a wide range of UQ methods.

Table 11 reports area under the receiver operating characteristic (AUROC) scores under the standard setting, complementing the AUPR results presented in the main text. These results further confirm that MoDEX consistently achieves strong performance across different datasets and evaluation metrics.

Table 12 presents comprehensive results for distribution shift detection using CIFAR-10 as the ID dataset. In addition to the result in the main text, which reports results for severity levels $\mathcal{C} \in \{1, 3, 5\}$, this table provides the full results for $\mathcal{C} \in \{1, 2, 3, 4, 5\}$. MoDEX consistently outperforms the baseline methods across all corruption severity levels, highlighting its effectiveness in detecting a wide range of distribution shifts. Table 13 reports the same distribution shift detection result evaluated using AUROC. Consistent trends are observed, indicating that MoDEX maintains robust performance regardless of the evaluation metric.

**Visualization on CIFAR-10-C.** We further provide qualitative visualizations on CIFAR-10-C to examine whether the epistemic uncertainty of MoDEX changes consistently with corruption severity. Following the setting used for distribution-shift detection, we evaluate MoDEX on corrupted CIFAR-10 samples with severity levels $\mathcal{C} = 1, 3, 5$. As shown in Figure 2, epistemic uncertainty increases as the corruption level becomes larger. This behavior is consistent with the expected property of a reliable uncertainty estimator, since more severe corruptions induce stronger distribution shifts from the clean CIFAR-10 training distribution.

*Table 14.* UQ-related downstream task results on CIFAR-10-LT under mild class imbalance ($\rho = 0.1$).

| Method | Test.Acc. | Miscl. AUPR. | SVHN / C-100 |
|---|---|---|---|
| Dropout | 70.87 ±3.0 | 89.82 ±2.3 | 37.37 ±1.4 / 61.18 ±1.3 |
| EDL | 79.09 ±0.4 | 95.36 ±0.1 | 72.18 ±2.1 / 80.09 ±0.7 |
| $\mathcal{I}$-EDL | 84.86 ±0.1 | 97.31 ±0.2 | 79.83 ±3.9 / 83.50 ±0.4 |
| R-EDL | 85.35 ±0.2 | 94.35 ±0.2 | 60.58 ±5.0 / 69.53 ±1.6 |
| DAEDL | 84.95 ±0.4 | 95.22 ±0.4 | 69.40 ±4.5 / 74.56 ±1.7 |
| Re-EDL | 84.43 ±0.8 | 94.75 ±0.7 | 59.08 ±10.9 / 72.37 ±3.5 |
| $\mathcal{F}$-EDL | 85.46 ±0.2 | 97.60 ±0.1 | 85.36 ±1.5 / 83.64 ±0.7 |
| **MoDEX** | **86.28** ±0.5 | **97.78** ±0.1 | **86.60** ±0.7 / **84.22** ±0.2 |

*Table 15.* AUROC scores for OOD detection using epistemic uncertainty estimates in the long-tailed setting. Results are reported with CIFAR-10-LT as the ID dataset under mild ($\rho = 0.1$) and severe ($\rho = 0.01$) imbalance, with SVHN and CIFAR-100 as the OOD datasets.

| Method | ID: CIFAR-10-LT ($\rho = 0.1$) OOD: SVHN / CIFAR-100 | ID: CIFAR-10-LT ($\rho = 0.01$) OOD: SVHN / CIFAR-100 |
|---|---|---|
| EDL | 80.36 ±1.5 / 77.03 ±0.6 | 58.25 ±4.4 / 61.05 ±0.7 |
| $\mathcal{I}$-EDL | 84.65 ±3.8 / 80.83 ±0.5 | 61.47 ±7.4 / 65.12 ±1.4 |
| R-EDL | 77.39 ±5.3 / 72.12 ±0.9 | 62.24 ±2.9 / 64.03 ±0.6 |
| DAEDL | 80.34 ±2.9 / 73.92 ±1.3 | 64.21 ±4.3 / 63.49 ±0.6 |
| Re-EDL | 70.24 ±7.9 / 71.85 ±3.7 | 46.19 ±11.0 / 57.76 ±2.3 |
| $\mathcal{F}$-EDL | 89.22 ±1.7 / 81.54 ±0.6 | 71.62 ±1.9 / 67.74±1.8 |
| **MoDEX** | **90.50** ±0.6 / **82.58** ±0.3 | **79.34** ±3.2 / **74.05** ±0.7 |

## F.2. Long-Tailed Setting

Table 14 reports UQ-related downstream task results under a mildly imbalanced setting using the CIFAR-10-LT dataset with mild imbalance ($\rho = 0.1$). These results demonstrate that MoDEX consistently exhibits robust UQ performance under class imbalance, complementing the severe imbalance results presented in the main text.

Table 15 presents AUROC scores for OOD detection under the long-tailed setting, evaluated on CIFAR-10-LT with imbalance factors $\rho = 0.1$ and $\rho = 0.01$. Consistent with the AUPR results, MoDEX consistently outperforms competing methods across both evaluation metrics.

*Table 16.* Comparison with recent non-EDL single-pass UQ methods using DMNIST-LT ($\rho = 0.01$) as the ID dataset.

| Method | Test.Acc. | Miscl. AUPR. | FMNIST |
|---|---|---|---|
| DDU | 54.47 $_{\pm 0.5}$ | 79.44 $_{\pm 2.2}$ | 98.77 $_{\pm 0.4}$ |
| Density-Softmax | 49.42 $_{\pm 1.5}$ | 79.23 $_{\pm 1.6}$ | 96.97 $_{\pm 1.1}$ |
| **MoDEX** | **69.42** $_{\pm 1.3}$ | **88.76** $_{\pm 1.0}$ | **98.88** $_{\pm 0.2}$ |

*Table 17.* Ablation study results on MoDEX parameters using CIFAR-10 as the ID dataset. "Fix-$\boldsymbol{\omega}$" fixes $\boldsymbol{\omega}(\mathbf{x}^\star)$ to a uniform vector, i.e., $\boldsymbol{\omega}(\mathbf{x}^\star) = \mathbf{1}/K$; "Fix-$\boldsymbol{\tau}$" sets $\tau_k(\mathbf{x}^\star)$ to be equal for all classes, i.e., $\tau_k(\mathbf{x}^\star) = \tau(\mathbf{x}^\star), \forall k \in [K]$; "Fix-$\boldsymbol{\omega}$ & Fix-$\boldsymbol{\tau}$" fixes both $\boldsymbol{\omega}$ and $\boldsymbol{\tau}$.

| Variant | Test.Acc. | Miscl. AUPR. | SVHN / C-100 |
|---|---|---|---|
| Fix-$\boldsymbol{\omega}$ & Fix-$\boldsymbol{\tau}$ | 91.67 $_{\pm 0.1}$ | 98.65 $_{\pm 0.0}$ | 84.12 $_{\pm 4.6}$ / 79.90 $_{\pm 3.1}$ |
| Fix-$\boldsymbol{\omega}$ | 90.83 $_{\pm 0.4}$ | 98.89 $_{\pm 0.8}$ | 87.13 $_{\pm 2.8}$ / 87.21 $_{\pm 0.5}$ |
| Fix-$\boldsymbol{\tau}$ | 91.53 $_{\pm 0.1}$ | 98.74 $_{\pm 0.2}$ | 89.85 $_{\pm 2.3}$ / 87.72 $_{\pm 0.3}$ |
| **MoDEX** | **92.46** $_{\pm 0.2}$ | **99.18** $_{\pm 0.0}$ | **91.58** $_{\pm 0.4}$ / **89.28** $_{\pm 0.3}$ |

*Table 18.* Ablation study on the regularization terms in MoDEX using CIFAR-10 as the ID dataset. "w/o $\boldsymbol{\omega}$-reg" denotes training without the regularization term on $\boldsymbol{\omega}$, "w/o $\boldsymbol{\tau}$-reg" refers to training without the regularization term on $\boldsymbol{\tau}$, and "w/o $\boldsymbol{\tau}$-reg & $\boldsymbol{\omega}$-reg" indicates training without both regularization terms.

| Variant | Test.Acc. | Miscl. AUPR. | SVHN / C-100 |
|---|---|---|---|
| w/o $\boldsymbol{\tau}$-reg & $\boldsymbol{\omega}$-reg | 91.63 $_{\pm 0.9}$ | 98.71 $_{\pm 0.2}$ | 79.52 $_{\pm 11.4}$ / 81.13 $_{\pm 0.6}$ |
| w/o $\boldsymbol{\tau}$-reg | 92.25 $_{\pm 0.2}$ | 99.13 $_{\pm 0.1}$ | 89.43 $_{\pm 1.0}$ / 87.09 $_{\pm 0.9}$ |
| w/o $\boldsymbol{\omega}$-reg | 91.68 $_{\pm 0.8}$ | 98.88 $_{\pm 0.4}$ | 85.42 $_{\pm 8.0}$ / 85.16 $_{\pm 5.5}$ |
| **MoDEX** | **92.46** $_{\pm 0.2}$ | **99.18** $_{\pm 0.0}$ | **91.58** $_{\pm 0.4}$ / **89.28** $_{\pm 0.3}$ |

### F.3. Long-Tailed and Noisy Setting

We provide an additional comparison in the long-tailed and noisy setting using the DMNIST-LT dataset. In addition to the EDL-based baselines reported in the main text, we compare MoDEX with recent non-EDL single-pass UQ methods, including DDU (Mukhoti et al., 2023) and Density-Softmax (Bui & Liu, 2024). These methods estimate uncertainty through feature-space density-based mechanisms and therefore provide a complementary baseline to EDL approaches.

As shown in Table 16, MoDEX consistently outperforms DDU and Density-Softmax across classification accuracy, misclassification detection, and OOD detection. Given that feature-space density-based methods are particularly well-suited to relatively low-dimensional MNIST-style benchmarks, these results suggest that the gains of MoDEX are not merely due to increased model complexity but also stem from its structured evidence decomposition and uncertainty aggregation mechanism.

### F.4. Ablation Study

Table 17 presents the results of an ablation study on MoDEX parameters using the CIFAR-10 dataset. Specifically, we investigate the impact of the additional parameters, $\boldsymbol{\omega}$ and $\boldsymbol{\tau}$. We evaluate models where $\boldsymbol{\omega}(\mathbf{x}^\star)$ is fixed to a uniform vector and $\boldsymbol{\tau}(\mathbf{x}^\star)$ is fixed such that $\tau_k(\mathbf{x}^\star) = \tau(\mathbf{x}^\star)$ for all $k \in [K]$, with both parameters being fixed. The resulting variants show a notable decline in both prediction accuracy and UQ performance. In contrast, MoDEX, which jointly learns all parameters, achieves the best performance. These results underscore the synergistic effect of the courtroom parameters in enhancing the model's robustness.

Table 18 reports the results of an ablation study on the regularization terms of the MoDEX objective function using the CIFAR-10 dataset. Specifically, we compare models without regularization on $\boldsymbol{\omega}$, without regularization on $\boldsymbol{\tau}$, and without regularization on both parameters. The variants without regularization exhibit a significant performance drop, particularly in OOD detection. This degradation is likely due to the degenerate assignment of $\boldsymbol{\omega}$ and $\boldsymbol{\tau}$, which adversely affects the overall model performance. Additionally, the high standard deviations observed in the results without regularization on both terms suggest unstable training dynamics. These findings emphasize the importance of the regularization terms for $\boldsymbol{\omega}$ and $\boldsymbol{\tau}$ in

enabling MoDEX to achieve stable and optimal performance.

### F.5. Comparison with Deep Ensembles

*Table 19.* Comparison with Deep Ensembles using CIFAR-10 as the ID dataset. Deep Ensembles uses 5 independently trained models, whereas MoDEX requires only a single forward pass.

| Method | Test.Acc. | Miscl. AUPR. | SVHN | C-100 |
|---|---|---|---|---|
| Deep Ensembles ($M = 5$) | **92.55** $\pm 0.1$ | **99.32** $\pm 0.0$ | 84.77 $\pm 1.4$ | 89.15 $\pm 0.2$ |
| **MoDEX** | 92.46 $\pm 0.2$ | 99.18 $\pm 0.0$ | **91.58** $\pm 0.4$ | **89.28** $\pm 0.3$ |

We additionally compare MoDEX with Deep Ensembles (Lakshminarayanan et al., 2017), a strong sampling-based ensemble baseline for uncertainty estimation. This comparison is particularly relevant because MoDEX admits an ensemble-like interpretation, as discussed in Proposition 5.2, where its predictive distribution can be viewed as a weighted mixture of class-specific EDL experts. However, MoDEX differs fundamentally from Deep Ensembles: while Deep Ensembles rely on multiple independently trained models, MoDEX represents multiple expert opinions within a single structured model and requires only a single forward pass at inference.

As shown in Table 19, Deep Ensembles achieve slightly higher test accuracy and misclassification detection performance. In contrast, MoDEX achieves stronger OOD detection performance, especially on SVHN, while using only a single model. Together with the inference-time comparison in Table 9, these results indicate that MoDEX provides a favorable efficiency–UQ trade-off: it retains an ensemble-like inductive bias through its structured mixture formulation, while avoiding the computational cost of maintaining and evaluating multiple independently trained networks.

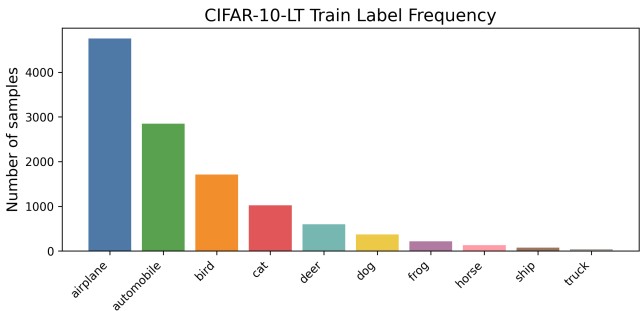

*Figure 3.* Class-wise label frequency of training set of CIFAR-10-LT with $\rho = 0.01$.

## G. Supplementary Analysis for Qualitative Evaluation

This section provides supplementary analysis to the qualitative study on semantically interpretable, uncertainty-aware classification presented in the main text. We begin by discussing how uncertainty-aware classification, framed through the courtroom analogy, can be interpreted, with a particular focus on ambiguous tail-class inputs. Additionally, we present further qualitative examples that complement the results shown in Figure 1.

### G.1. Interpreting Uncertainty-Aware Classification under the Courtroom Analogy

**Setting.** We train the models on the CIFAR-10-LT dataset with $\rho = 0.01$, and the class-wise label frequencies are shown in Figure 3. Our focus is on the uncertainty-aware classification for the challenging test input $\mathbf{x}^\star$, where the ground-truth label belongs to a tail class, but exhibits *semantic ambiguity* with one or more visually similar head classes. To operationalize this ambiguity, we select examples with high entropy in the human-annotated label distribution $\boldsymbol{\mu}_{\text{human}}(\mathbf{x}^\star)$ (from CIFAR-10H), i.e., $\mathbb{H}[\boldsymbol{\mu}_{\text{human}}(\mathbf{x}^\star)] = -\sum_{k=1}^{K} \mu_{\text{human},k}(\mathbf{x}^\star) \log \mu_{\text{human},k}(\mathbf{x}^\star)$.

Such *ambiguous tail* cases represent one of the most failure-prone scenarios for uncertainty-aware classification in the long-tailed setting, since a model must simultaneously (i) avoid collapsing to a frequent head class due to imbalance and (ii) avoid overconfidently committing to a single label when the input itself is inherently ambiguous.

We do not claim that there exists a unique or ideal parameter configuration for each input, nor that MoDEX necessarily recovers an *optimal* uncertainty representation. Instead, our emphasis is that MoDEX enables a *principled and interpretable* uncertainty aggregation mechanism, where its prediction can be explained through the behavior of the courtroom parameters $(\boldsymbol{\alpha}(\mathbf{x}^\star), \boldsymbol{\tau}(\mathbf{x}^\star), \boldsymbol{\omega}(\mathbf{x}^\star))$, revealing *why* a certain class wins and *where* ambiguity or imbalance-driven effects are expressed.

**How to read the courtroom parameters.** MoDEX predicts $K$ advocate opinions, each modeled as a Dirichlet distribution with concentration vector $\boldsymbol{\alpha}_k(\mathbf{x}^\star) = \boldsymbol{\alpha}(\mathbf{x}^\star) + \tau_k(\mathbf{x}^\star)\mathbf{e}_k$. To recap, under this structured decomposition, $\boldsymbol{\alpha}(\mathbf{x}^\star)$ acts as *shared evidence* (the objective facts of the case acknowledged by all advocates), while $\tau_k(\mathbf{x}^\star)$ acts as *class-specific advocacy strength* (how strongly advocate $k$ pushes its own verdict beyond the shared evidence). Finally, $\boldsymbol{\omega}(\mathbf{x}^\star)$ represents *advocacy plausibility* (the judge's assessment of how persuasive each advocate is for this particular case). The aggregated predictive mean $\boldsymbol{\mu}(\mathbf{x}^\star)$ is then obtained by deliberating over these opinions via $\boldsymbol{\omega}(\mathbf{x}^\star)$.

For clean and unambiguous ID inputs, we typically observe a near-consensus court: $\boldsymbol{\omega}(\mathbf{x}^\star)$ is sharply peaked, and the corresponding advocate exhibits large $\tau_k(\mathbf{x}^\star)$ together with concentrated shared evidence, yielding a confident prediction. In contrast, for ambiguous ID inputs—especially ambiguous tail inputs—MoDEX often assigns non-trivial plausibility to multiple advocates, and the resulting prediction becomes interpretable through *which channel* encodes which effect.

**Ambiguous tail inputs: two recurring aggregation patterns.** Empirically, MoDEX's uncertainty aggregation behavior for ambiguous tail inputs often falls into two recurring and interpretable patterns. These are not meant to be exhaustive, but they provide a useful lens for interpreting the qualitative examples.

- **(i) Tail-aligned deliberation under ambiguity.** In this pattern, MoDEX reaches a tail-consistent verdict through a coherent alignment of the courtroom components toward the tail label, while still retaining uncertainty due to semantic ambiguity.

*Interpretation.* Even when the tail class is under-represented during training, the input $\mathbf{x}^\star$ may contain sufficiently distinctive visual evidence. In such cases, (a) the shared evidence $\boldsymbol{\alpha}(\mathbf{x}^\star)$ assigns relatively more mass to the tail label than to its confusable head competitors; (b) the tail advocate attains non-negligible plausibility $\omega_{\text{tail}}(\mathbf{x}^\star)$; and (c) the tail advocate also argues decisively via a substantial $\tau_{\text{tail}}(\mathbf{x}^\star)$. These effects reinforce each other during deliberation, resulting in a predictive mean $\boldsymbol{\mu}(\mathbf{x}^\star)$ that is aligned with the tail label.

Importantly, the resulting predictions are typically not overly confident: $\boldsymbol{\omega}(\mathbf{x}^\star)$ often remains non-degenerate, $\tau_{\text{head}}(\mathbf{x}^\star)$ and $\tau_{\text{tail}}(\mathbf{x}^\star)$ are often competitive, and the base evidence for the head class is non-negligible. Together, these factors enable the model to represent ambiguity while still achieving correct predictions.

- **(ii) Separating class imbalance from shared evidence.** In this pattern, MoDEX explicitly separates imbalance-induced class-specific effects from the shared-evidence channel. The decomposition $\boldsymbol{\alpha}_k(\mathbf{x}^\star) = \boldsymbol{\alpha}(\mathbf{x}^\star) + \tau_k(\mathbf{x}^\star)\mathbf{e}_k$ provides a dedicated pathway for class-wise asymmetries via $\boldsymbol{\tau}$ without overloading the shared evidence $\boldsymbol{\alpha}$.

  *Interpretation.* The shared evidence $\boldsymbol{\alpha}(\mathbf{x}^\star)$ can remain relatively case-faithful, attributed to the objective cues. while head-class advocates may exhibit stronger advocacy strengths $\tau_k(\mathbf{x}^\star)$ due to frequent exposure during training. Crucially, this imbalance-induced effect is expressed primarily through the advocacy channel, rather than forcing the shared evidence itself to collapse toward head classes. Moreover, $\boldsymbol{\omega}(\mathbf{x}^\star)$ often assigns substantial plausibility to multiple advocates. The final predictive mean $\boldsymbol{\mu}(\mathbf{x}^\star)$ therefore emerges as an interpretable compromise that is both robust to imbalance and faithful to semantic ambiguity.

**Takeaway.** The two mechanisms above illustrate why MoDEX can be both robust and interpretable in the ambiguous tail regime: (i) it can produce a tail-consistent verdict through coherent tail-aligned deliberation, and (ii) it can isolate imbalance-induced class-specific effects in the advocacy channel, keeping the shared evidence channel less distorted. In both cases, inspecting $(\boldsymbol{\alpha}(\mathbf{x}^\star), \boldsymbol{\tau}(\mathbf{x}^\star), \boldsymbol{\omega}(\mathbf{x}^\star))$ provides a concrete diagnostic for *where* the model's confidence and uncertainty originate.

## G.2. Case Studies for Uncertainty-Aware Classification

We present additional case studies for uncertainty-aware classification, comparing EDL and MoDEX, to supplement the results shown in Figure 1.

**Figure 4.** This input is clearly ID with ground-truth class *dog*. EDL collapses to *cat*, a relatively frequent head-class with more training samples. In contrast, both human annotators and MoDEX correctly predict the label with high confidence, reflecting low semantic ambiguity. This example serves as a sanity check, showing that MoDEX behaves consistently with standard models when uncertainty is minimal.

**Figure 5.** This input corresponds to *dog* but exhibits ambiguity with multiple classes, including *bird* and *frog*. As shown in the CIFAR-10H annotation distribution, the image displays high human disagreement and elevated annotation entropy. While EDL correctly predicts *dog*, it remains overly confident and fails to capture this inherent ambiguity. In contrast, MoDEX assigns high base evidence to *dog*, and shows strong advocacy for multiple classes including *dog*, *bird*, and *frog*. It also assigns substantial plausibility weights to multiple classes, reflecting the image's ambiguity. The aggregated parameters produce the prediction $\boldsymbol{\mu}(\mathbf{x}^\star)$, which closely aligns with $\boldsymbol{\mu}_{\text{human}}(\mathbf{x}^\star)$, indicating that MoDEX accurately captured the semantic ambiguity.

**Figure 6.** This input is labeled as *dog* but is semantically ambiguous with *cat*, which is a relatively head class under the long-tailed training distribution. EDL again collapses its prediction toward *cat*, yielding an overconfident and incorrect outcome. In contrast, MoDEX provides a robust uncertainty-aware prediction by assigning strong support to *dog* across its parameters while remaining resilient to head-class bias. Consequently, the predicted mean $\boldsymbol{\mu}(\mathbf{x}^\star)$ closely matches the human annotation distribution $\boldsymbol{\mu}_{\text{human}}(\mathbf{x}^\star)$.

**Figure 7.** This image corresponds to the class *horse*. While the correct label is obvious to human annotators due to contextual cues such as riding, it poses challenges for standard models. The EDL model incorrectly predicts *deer* with high confidence, failing to provide a robust prediction and overlooking the semantic similarity between the two classes. In contrast, MoDEX correctly predicts *horse* while explicitly capturing its ambiguity with *deer*, reflecting a more calibrated and semantically meaningful uncertainty representation.

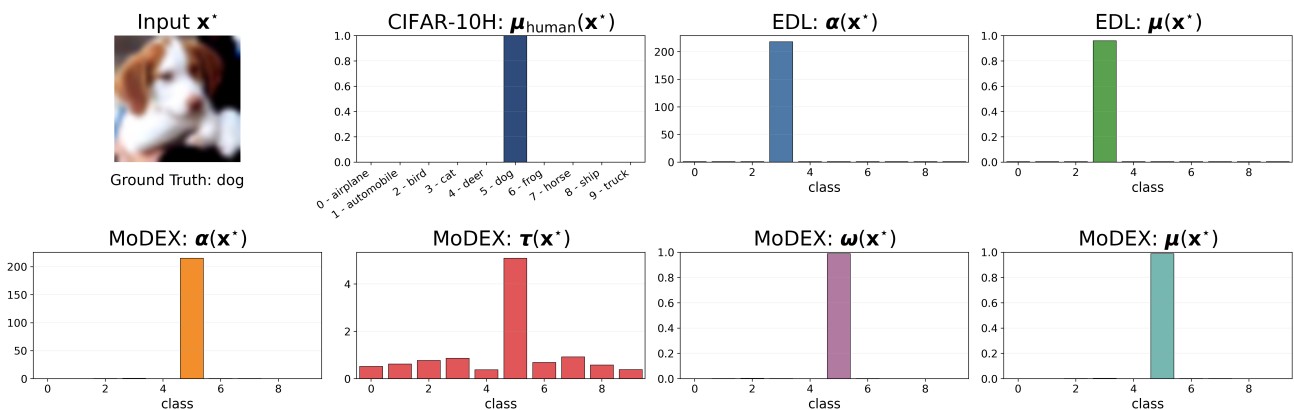

*Figure 4.* Uncertainty-aware classification results for EDL and MoDEX trained on CIFAR-10-LT, evaluated on a clearly in-distribution test input $\mathbf{x}^\star$ with ground-truth class *dog*.

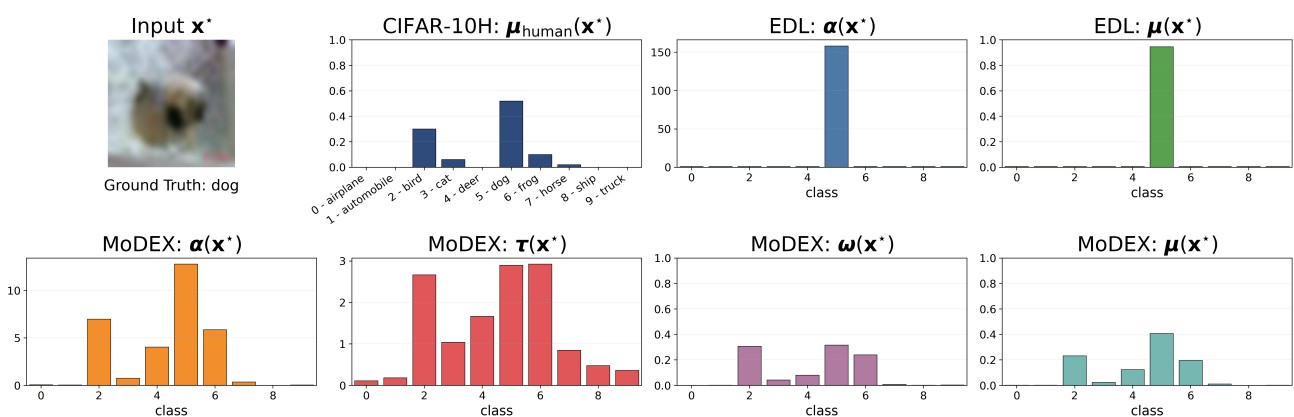

*Figure 5.* Uncertainty-aware classification results for EDL and MoDEX trained on CIFAR-10-LT, evaluated on a test input $\mathbf{x}^\star$ with ground-truth class *dog* and semantic ambiguity with *bird* and *frog*.

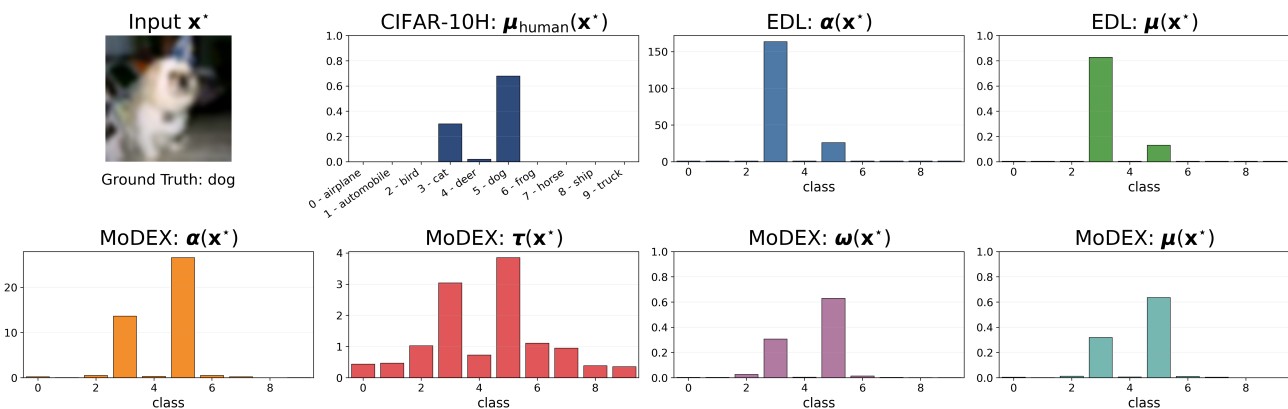

*Figure 6.* Uncertainty-aware classification results for EDL and MoDEX trained on CIFAR-10-LT, evaluated on a test input $\mathbf{x}^\star$ with ground-truth class *dog* and semantic ambiguity with the head class *cat*.

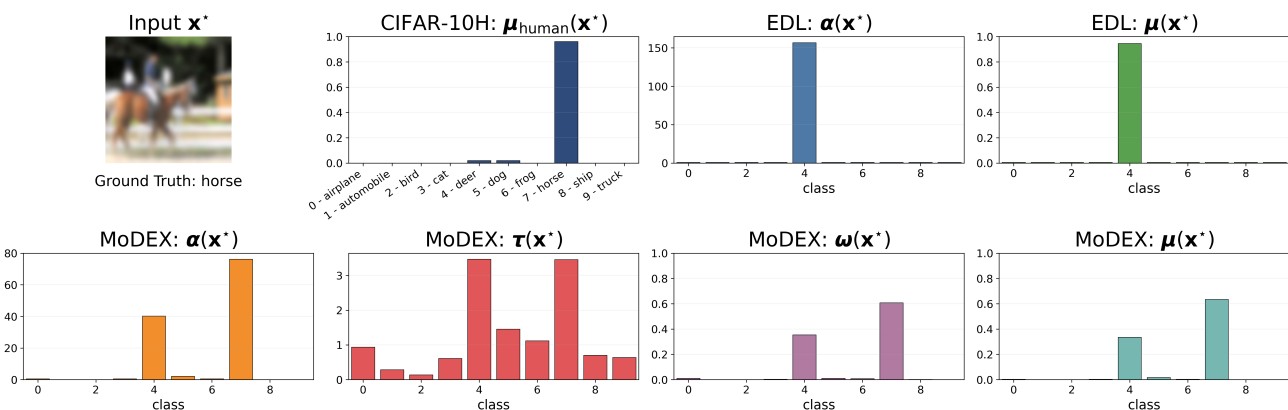

*Figure 7.* Uncertainty-aware classification results for EDL and MoDEX trained on CIFAR-10-LT, evaluated on a test input $\mathbf{x}^\star$ with ground-truth class *horse*.

## H. Additional Qualitative Study: Decreasing Epistemic Uncertainty

We provide an additional qualitative study to examine whether the epistemic uncertainty induced by MoDEX exhibits a desirable data-dependent behavior. Bengs et al. (2022) formalized a desideratum for epistemic uncertainty, requiring that epistemic uncertainty decrease as more training data are observed. Subsequent studies (Shen et al., 2024; Yoon & Kim, 2026) have operationalized this property by evaluating whether the estimated epistemic uncertainty decreases as the training set size increases. These studies show that standard EDL-style epistemic uncertainty can fail to exhibit this desired vanishing behavior, motivating us to examine whether MoDEX satisfies the same qualitative criterion.

Following this evaluation protocol, we train MoDEX on progressively larger subsets of the CIFAR-10 training set and evaluate its epistemic uncertainty on a fixed CIFAR-10 test set. Specifically, we use VGG-16 as the backbone and train MoDEX with subset sizes $|\mathcal{D}| \in \{500, 1000, 2000, 5000, 10000, 25000, 50000\}$. For all subset sizes, the evaluation set is kept fixed, and we report both test accuracy and average epistemic uncertainty on the held-out CIFAR-10 test set. This setup isolates the effect of increasing the amount of training data on the epistemic uncertainty estimate.

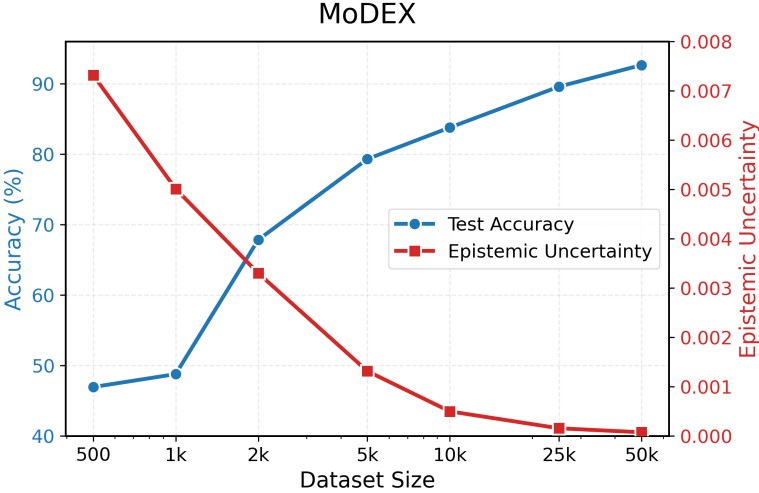

*Figure 8.* Qualitative evaluation of the decreasing epistemic uncertainty desideratum on CIFAR-10. Test accuracy (blue) and epistemic uncertainty (red) of MoDEX are evaluated across increasing training subset sizes. Accuracy is plotted on the left y-axis, and epistemic uncertainty is plotted on the right y-axis.

As shown in Figure 8, the epistemic uncertainty of MoDEX decreases as the number of training samples increases. This trend is consistent with the desideratum formalized by Bengs et al. (2022): as more observations from the underlying data distribution become available, the model's uncertainty over plausible class-probability vectors should be reduced. Thus, the

result provides qualitative evidence that MoDEX induces epistemic uncertainty with meaningful data-dependent behavior.

We emphasize that this experiment does not imply that MoDEX recovers a uniquely correct second-order predictive distribution from input-label supervision alone. Rather, it shows that the epistemic uncertainty induced by the courtroom-based aggregation mechanism behaves consistently with an important epistemic uncertainty desideratum under the qualitative protocol used in prior work (Shen et al., 2024; Yoon & Kim, 2026). This supports our view that, although MoDEX shares the general limitations of EDL-style methods regarding fully supervised epistemic uncertainty, its induced epistemic uncertainty remains practically meaningful for relative uncertainty assessment and downstream UQ tasks.

