# OpenReview forum: "Courtroom Analogy: New Perspective on Uncertainty-Aware Classification"
_ICML.cc/2026/Conference — ICML 2026 regular_

### Official Review · Reviewer_LpiM · 2026-03-10

**Soundness:** 3
**Presentation:** 4
**Significance:** 2
**Originality:** 2
**Overall Recommendation:** 4
**Confidence:** 4

**Summary:**

This paper proposes an EDL extension by introducing an extra one-hot-encoded parameter to the concentration parameter in the EDL setting for each class, and then aggregating them as a convex mixture. They provided a courtroom analogy for motivating such development. Experiments are done on representative enough datasets, on tasks that genuinely reflect characteristics of their quantified (aleatoric and epistemic) uncertainties.

**Compliance With Llm Reviewing Policy:**

Affirmed.

**Final Justification:**

The author has carefully articulated my concerns and provided convincing arguments for some fundamental limitations of EDL. I decided to increase my score from weak reject to weak accept to reflect this.

**Key Questions For Authors:**

1. Could you address the fundamental limitations that there exists no prinicpled objective functions of EDL in your work? How does your work go around that?

2. In terms of empirical performance, can you explain further why looking at aleatoric uncertainty of misclassified samples is a good metric?

**Limitations:**

Yes

**Strengths And Weaknesses:**

### Soundness
#### Strength:
- A coherent probabilistic formulation starting from an easy-to-understand hierarchical data-generating process.
- The method is really simple to understand.

#### Weakness:
1. EDL Fundamental Weakness is not addressed
- A potential concern relates to the theoretical limitations of evidential deep learning (EDL). Prior work by Bengs et al. shows that EDL-type models are not optimisable in a principled sense when only 0th-order supervision (input–label pairs) is available, as there exists no valid second-order proper scoring rule that uniquely identifies an optimal second-order predictive distribution in this setting. In other words, epistemic uncertainty cannot be learned in a fully principled way from such supervision alone. This limitation is less problematic for approaches such as Bayesian neural networks or deep ensembles, where epistemic uncertainty arises naturally from mechanisms such as weight priors or stochastic initialization across ensemble members. These sources of epistemic uncertainty are then updated during training. In contrast, EDL-based methods lack an explicit mechanism that justifies the origin of epistemic uncertainty within the model. Given that the proposed method builds upon the EDL framework, it would be important for the paper to explicitly discuss the source and interpretation of epistemic uncertainty in this model. In particular, the authors should clarify why the proposed training objective is principled in light of the above limitation, and how the framework avoids or mitigates the theoretical concerns raised in prior work.

2. Empirical evaluations are slightly confusing to me
- It makes sense to use OOD detection for assessing epistemic uncertainty quality, but it is confusing to me that you check whether aleatoric uncertainty is higher for misclassification samples than for correct ones. This alone seems to assume all errors made are because there are no epistemic uncertainty? I am happy to be convinced but I think this evaluation metric is a bit ill-posed.

### Presentation
#### Positive
1. Well written and easy to understand. The courtroom analogy provides an intuitive explanation of the modelling framework.

#### Weakness
1. Technical jargons like subjective logic and Demster-Shafer's theory have been thrown out without diving deeper, or at least given an intuitive explanations of what they are doing. I know it is not this paper's responsibility to do this but I felt intimidated as a reader seeing these technical terms without explanations, I felt like "Am I supposed to be convinced about this work because now I see these unexplained terms?"

### Significance
#### Positive
1. Single pass uncertainty-aware modelling for deep learning is an important problem. By incorporating more flexible parameters in the model, the method proposed MoDEX performs better than other EDL variants.
#### Negative
1. The contribution is quite moderate because it builds on the EDL framework which are known to be a fundamentally ill-posed problem...

### Originality
It is to the best of my knowledge, a new approach to EDL, but the components used are quite moderate. Doing a mixture/ensemble of Dirichilet weights is not really that new technically...

---

> ### Author Rebuttal · Authors · 2026-03-31
>
> Thank you for your careful review of our paper and for the insightful and constructive comments. Please find our detailed answers to your comment below.
>
> **1. (How does MoDEX address the theoretical limitations of EDL, and why is it still meaningful?)**
>
> **(How we position MoDEX relative to prior critiques.)** We agree that prior works identified an important limitation of learning second-order predictive distributions from only input–label pairs. However, we believe this limitation should be interpreted in light of what MoDEX is intended to do.
>
> First, MoDEX is **not intended to recover a fixed but unknown ground-truth second-order target** from input-label pairs via proper scoring rules. Rather, it is an uncertainty-aware model whose epistemic uncertainty is **induced by the modeling mechanism itself**, with labels providing only directional guidance.
> Specifically, this uncertainty arises from the courtroom-based aggregation of expert-wise opinions, capturing two sources:  (i) disagreement across experts and (ii) imprecision within each expert. In this sense, MoDEX is closer in spirit to deterministic UQ methods, where uncertainty is induced by modeling assumptions or inductive biases rather than directly supervised.
>
> Second, **prior critiques [1–2] should not be treated as the sole standard** for evaluating later EDL methods, including MoDEX. The non-existence of a *proper second-order scoring rule* does not imply that epistemic uncertainty is meaningless; rather, it only rules out one justification via proper-scoring-rule elicitation. Indeed, follow-up work [3] has already explored alternative criteria, such as coherence with the *reference second-order distribution*.
> At the formal level, these critiques also do not directly cover MoDEX. In particular, [1-2] analyze specific loss formulations, whereas our objective—especially the regularization term—falls outside that scope. In addition, the conditions (i) and (ii) in Thm 5.1 of [2] are explicitly non-exhaustive, so direct coverage of a more structured model like MoDEX is nontrivial.
>
> Third, we still take this concern seriously and have added a qualitative study on CIFAR-10 showing that MoDEX satisfies the **desideratum in Def. 1-(A1) of [1]**: epistemic uncertainty **decreases as more data are observed** (see https://anonymous.4open.science/r/ICML2026_11275-D6EA/Fig2.pdf).
>
> **(Why the contribution remains meaningful.)** We believe that the theoretical limitations identified for EDL in prior work do not diminish the contribution of MoDEX. Existing critiques mainly challenge the **quantitative or absolute interpretations** of epistemic uncertainty in this framework, rather than its practical usefulness [3]. In many downstream settings, however, **what matters more is relative epistemic uncertainty**—e.g., whether one example should be treated as more uncertain than another.
> From this perspective, the value of a UQ method lies not only in whether it recovers a uniquely correct epistemic target, but also in providing a practical, efficient, and interpretable framework, which is why we believe MoDEX remains meaningful.
>
> **2. (Is aleatoric uncertainty suitable for misclassification detection?)**
>
> We use **aleatoric uncertainty** because it is defined on the **first-order predictive distribution**, which directly determines the final prediction and its correctness. This does not mean that all misclassifications are purely aleatoric; rather, we evaluate whether the uncertainty in the final predictive distribution increases when the prediction is incorrect. This is also consistent with MoDEX, where aleatoric uncertainty is defined after aggregating experts’ opinions, whereas epistemic uncertainty is defined on the underlying second-order belief structure.
>
> This choice is also reasonable because, even in a well-generalized ID setting, many misclassifications arise from irreducible **intrinsic data ambiguity** rather than model uncertainty. As shown in Fig. 1 of our paper, some CIFAR-10 images are inherently ambiguous, as reflected in $\mu_{human}(x)$ in CIFAR-10H. In our framework, such inputs yield a *diffuse* first-order predictive distribution and thus high aleatoric uncertainty.
>
> Although epistemic or total uncertainty could be considered, we focus on aleatoric uncertainty because it is the most direct measure of ambiguity in the final prediction, whereas epistemic uncertainty is more naturally suited to OOD detection.
>
> **3. (SL/DST are introduced with limited intuition.)**
>
> Terms such as subjective logic and Dempster-Shafer Theory are included mainly as background to motivate the use of Dirichlet-based opinions and to situate the work within the EDL literature, not as prerequisites for understanding the main contribution. In the revised version, we will add intuitive explanations and clarify their role in our formulation.
>
> **References**
>
> [1] Bengs et al., arXiv:2203.06102
>
> [2] Bengs et al., arXiv:2301.12736
>
> [3] Jürgens et al., arXiv:2402.09056v3

---

> > ### Author Rebuttal · Reviewer_LpiM · 2026-04-01
> >
> > The author has carefully articulated my concerns and provided convincing arguments for some fundamental limitations of EDL. I decided to increase my score from weak reject to weak accept to reflect this.

---

> > > ### Author Response · Authors · 2026-04-01
> > >
> > > Dear Reviewer LpiM,
> > >
> > > Thank you very much for your thoughtful re-evaluation and for revising your score. We truly appreciate your careful and constructive feedback throughout the review process. Your comments have been extremely helpful in refining our arguments, particularly in sharpening our discussion on the limitations of EDL and how our work should be positioned relative to them.
> > >
> > > We are very glad that our responses have addressed your concerns.
> > >
> > > Best regards,
> > > Authors

---

### Official Review · Reviewer_4TUv · 2026-03-11

**Soundness:** 3
**Presentation:** 4
**Significance:** 3
**Originality:** 3
**Overall Recommendation:** 4
**Confidence:** 5

**Summary:**

The authors propose the Courtroom Analogy framework to model uncertainty perception classification as structured debates between K specific classes of defense counsel. The Mixture of Dirichlet Experts (MoDEX) model proposed based on this framework achieves efficient and interpretable uncertainty quantification under single pass forward by decomposing the concentration parameters of the Dirichlet distribution into shared evidence and class specific advocacy strength.

**Compliance With Llm Reviewing Policy:**

Affirmed.

**Final Justification:**

The primary value of this work lies in its excellent storytelling and the structural inductive bias it introduces to the UQ community. While theoretical novelty is largely an extension of existing Dirichlet-type families, the proposed decomposition offers a clear diagnostic tool for practitioners to understand why a model is uncertain. The manuscript is now complete, the experimental evidence is robust, and the logic is self-consistent. Because the work provides meaningful insights into uncertainty aggregation without sacrificing computational efficiency, I recommend a weak accept.

**Key Questions For Authors:**

Refer to weaknesses.

**Limitations:**

yes

**Strengths And Weaknesses:**

Strengths：
	1. The "courtroom analogy" is proposed as a highly intuitive and semantically clear theoretical framework for understanding predictive uncertainty in single-pass models.
	2. Decoupling the Dirichlet concentration parameters into shared evidence and class-specific advocacy strength is a novel design that effectively resolves the parameter entanglement issue found in standard evidential deep learning.
	3. The establishment of equivalence between MoDEX and the Extended Flexible Dirichlet (EFD) distribution provides a rigorous statistical foundation for the methodology.

Weaknesses：
	1. While various EDL variants were compared, there is a lack of in-depth comparison with the latest non-EDL-based single-pass quantification methods, such as other SOTA approaches based on feature-space density. This makes it difficult to determine whether MoDEX's leading performance stems from the structural design of the "courtroom analogy" or is simply due to its increased complexity relative to the base EDL model.
	2. Although the semantic subjectivity of courtroom analogy provides intuitive understanding, defining \alpha\left(x\right) as an "objective fact" and \tau_k\left(x\right) as a "subjective defense" is to some extent an artificially imposed semantics. How can neural networks ensure that these parameters evolve strictly according to the expected logic of a "judge" or "lawyer" in end-to-end training?
	3. The experts' plausibility weights \omega\left(x\right) are computed using a Softmax function. In OOD scenarios, Softmax is highly prone to producing overconfident predictions. If, in regions far from the training distribution, the Softmax function disproportionately skews the weights toward a single "ignorant" expert, would this mechanism undermine the overall reliability of the uncertainty quantification?

---

> ### Author Rebuttal · Authors · 2026-03-29
>
> Thank you for your careful review of our paper and for the insightful and constructive comments. Please find our detailed answers to your comment below.
>
>
> **1. (The comparison to recent non-EDL single-pass UQ methods is insufficient.)**
>
>
> We note that a representative non-EDL single-pass UQ baseline, **DUQ** (van Amersfoort et al., ICML 2020), is already included in Table 9 of our paper. To further strengthen this comparison, we newly add **DDU [1]** and **Density-Softmax [2]**, two strong feature-space density-based single-pass UQ baselines, on the DMNIST-LT benchmark. As shown in **Table 1**, **MoDEX consistently outperforms both methods**, even though feature-space density-based methods are known to be particularly effective in low-dimensional settings.
>
>
> This suggests that MoDEX’s performance gains are not merely due to increased model complexity, but to its structured mechanism for organizing and aggregating evidence, which yields more effective uncertainty-aware predictions without density-based components.
>
>
> **[Table 1. Comparison with single-pass UQ methods on DMNIST-LT]**
>
> |Method|Test.Acc. ↑|Miscl. AUPR. ↑|OOD AUPR. (FMNIST) ↑|
> |---|---:|---:|---:|
> | DDU |54.47 ± 0.5| 79.44 ± 2.2 |98.77 ± 0.4|
> | Density-Softmax |49.42 ± 1.5|79.23 ± 1.6|96.97 ± 1.1|
> | **MoDEX** | **69.42 ± 1.3** | **88.76 ± 1.0** | **98.88 ± 0.2** |
>
> We will include these results and report additional single-pass UQ comparisons in the revised appendix.
>
>
> **2. (Validity of the courtroom semantics under end-to-end training.)**
>
>
> We do not claim that end-to-end training can *strictly guarantee* a literal *judge/lawyer* semantics for $\alpha(x)$ and $\tau_{k}(x)$, since these latent roles are not directly supervised.  Rather, the courtroom analogy serves as a structured inductive bias for how uncertainty is parameterized and aggregated.
>
> Specifically, $\alpha(x)$ is shared across all class-wise experts, whereas $\tau_{k}(x)$ contributes only to the $k$-th expert through the class-specific term. While this does not *guarantee* a literal semantic decomposition, it does *encourage* $\alpha(x)$ to capture and consolidate evidence that is common across competing class-wise interpretations, and $\tau_{k}(x)$ to capture the additional class-specific emphasis beyond that shared component.
>
> Moreover, the qualitative analyses in Fig. 1 and Fig. 3-6 in our paper suggest that the learned courtroom parameters are broadly aligned with this interpretation in practice. That is, although they are not directly supervised as semantic variables, they exhibit patterns consistent with shared evidence, class-specific emphasis, and their input-dependent aggregation. Our qualitative section is intended to demonstrate this interpretable behavior in practice, rather than to claim uniquely identifiable semantic recovery.
>
> Thus, the analogy is valuable not for enforcing literal judge/lawyer roles, but for introducing a more structured and interpretable parameterization of uncertainty.
>
>
> **3. (Does softmax-based expert weighting lead to overconfident expert selection in OOD regions?)**
>
>
> We agree that a softmax-based gating network can raise concerns about concentrated expert allocation, especially in OOD regions. However, this does not directly imply unreliable UQ in our model.
>
>
> First, the concern is not the softmax function alone, but its typical coupling with cross-entropy (CE), which strongly favors near one-hot assignments. In contrast, the plausibility weights $\omega(x)$ in our framework are trained using a Brier-score-based objective rather than CE, yielding a substantially softer training signal and thus reducing the tendency toward overly concentrated expert allocations.
>
>
> Second, $\omega(x)$ is not trained as an independent classifier, but jointly optimized with other courtroom parameters under the full objective. Therefore, assigning most mass to one expert is preferred only if it improves overall evidential fit and uncertainty calibration.
>
>
> Third, even if $\omega(x)$ becomes sharp, this does not by itself yield overconfident predictions. In our framework, the predictive uncertainty is determined jointly by the plausibility weights and each expert’s own uncertainty. Therefore, even if one expert receives most of the plausibility mass, the prediction can remain appropriately uncertain when that expert itself is uncertain.
>
>
> Empirically, we do not observe pathological single-expert collapse in OOD or ambiguous regions. As shown in Fig. 1 and Fig. 3-6 in our paper, $\omega(x)$ remains distributed across multiple experts for ambiguous inputs, while becoming sharper mainly for confident ID samples. This suggests that the gating behaves as intended rather than collapsing to overconfident expert selection.
>
> **References**
>
> [1] Mukhoti et al., Deep Deterministic Uncertainty: A New Simple Baseline, CVPR 2023.
>
> [2] Bui and Liu, Density-Softmax: Efficient Test-time Model for Uncertainty Estimation and Robustness under Distribution Shifts, ICML 2024.

---

> > ### Author Rebuttal · Reviewer_4TUv · 2026-04-02
> >
> > Based on my overall evaluation of the manuscript, I will maintain my weak accept score.

---

> > > ### Author Response · Authors · 2026-04-04
> > >
> > > Dear Reviewer 4TUv,
> > >
> > > Thank you very much for maintaining your positive rating. We sincerely appreciate your thoughtful and constructive feedback throughout the review process.
> > >
> > > Your comments have been particularly valuable in helping us further strengthen the paper. In the revised version, we will incorporate additional comparisons with recent non-EDL single-pass baselines, refine the interpretation of the courtroom analogy to avoid overly imposed semantics, and provide a more precise discussion of the behavior of the softmax-based plausibility weights under OOD scenarios.
> > >
> > > We are very glad that our responses have addressed your concerns.
> > >
> > > Best regards,
> > >
> > > Authors

---

### Official Review · Reviewer_ivYU · 2026-03-12

**Soundness:** 3
**Presentation:** 3
**Significance:** 3
**Originality:** 2
**Overall Recommendation:** 4
**Confidence:** 4

**Summary:**

This paper introduces an innovative perspective on uncertainty-aware classification by drawing an analogy to courtroom deliberation. The authors propose that classification can be viewed as a structured debate among K advocates. Each advocate represents a class and forms a probabilistic opinion modeled as a Dirichlet distribution, with a structured decomposition into shared evidence and class-specific advocacy. These opinions are aggregated via input-dependent plausibility weights, yielding a mixture of Dirichlet distributions. The authors instantiate this framework as Mixture of Dirichlet Experts (MoDEX) by a neural architecture that predicts all courtroom parameters in a single forward pass. Theoretical contributions include proving MoDEX generalizes existing models based on EDL (Theorem 5.1), showing equivalence to both an ensemble of EDL experts (Proposition 5.2) and a mixture of base EDL and softmax predictors (Theorem 5.3), and decomposing epistemic uncertainty into inter-expert uncertainty and intra-expert uncertainty (Proposition 5.4). Extensive experiments on CIFAR, CIFAR-LT, DMNIST-LT, and CIFAR-10-C demonstrate state-of-the-art performance across multiple UQ tasks. In addition, qualitative analysis shows interpretable parameter behavior aligned with human uncertainty judgments.

**Compliance With Llm Reviewing Policy:**

Affirmed.

**Final Justification:**

This paper introduces a new perspective on uncertainty-aware classification through a courtroom analogy, formulating classification as a debate among class-specific advocates. The proposed Mixture of Dirichlet Experts (MoDEX) decomposes epistemic uncertainty into inter-expert and intra-expert components, providing improved interpretability. The framework is theoretically grounded, with proofs establishing its connections to existing evidential deep learning (EDL) methods, and demonstrates competitive performance across several benchmarks.

I acknowledge the clear structure, solid theoretical grounding, and interpretability benefits of the approach. However, the paper is perceived as having moderate originality, as it builds on existing EDL frameworks rather than introducing entirely new mathematical constructs. Additional concerns include limited computational cost analysis, insufficient diversity of evaluation metrics (e.g., OOD detection), and lack of deeper empirical exploration of uncertainty decomposition. After rebuttal, the authors have adequately addressed key concerns, particularly in originality.

Overall, I recommend acceptance. The paper offers a well-structured and interpretable framework that meaningfully extends existing methods, meeting the rigorous standards expected for this conference.

**Key Questions For Authors:**

Key questions for authors

1.Computational complexity: Could you provide the time comparisons between MoDEX and baseline(e.g. F-EDL, EDL) on the same hardware? MoDEX use three heads and its parameter is more than EDL so the time would help readers assess the additional overhead of MLP heads.

2.Metrics of OOD detection:  The experiments only use AUPR to evaluate the performance of OOD detection tasks. It seems to be simple. Could you think it is necessary to use more metrics such as FPR95?

3.Practical value of decomposition of epistemic uncertainty: Could you illustrate the practical significance of decomposition of epistemic uncertainty? For example, how can we do to reduce high inter-expert uncertainty or intra-expert uncertainty?

4.Generalization to other domains:  Obviously, the courtroom analogy is domain-agnostic. Have you considered apply MoDEX to non-image data to assess its generalization? What challenges might face?

**Limitations:**

Yes.

The authors mentioned that MoDEX may not fully capture all sources of uncertainty in highly complex, ambiguous, or adversarial environments, limited to classification and lacks explicit supervision on epistemic uncertainty.

**Strengths And Weaknesses:**

Strengths:

(1) The paper is sound. It provides complete mathematical derivations for all key components, including the generative process, structured decomposition, and uncertainty measures. The four theoretical results (Theorem 5.1, Proposition 5.2, Theorem 5.3, Proposition 5.4) form a coherent logical chain that establishes MoDEX as a principled generalization of existing methods.

(2) The paper is well-structured. It narrates clearly from problem statement and background to method and experiments. In addition, the courtroom analogy is introduced early and consistently used throughout, which makes complex concepts accessible.

(3) Existing UQ methods excel at quantifying uncertainty but provide little insight into its structure so that they are weak in interpretability. MoDEX can decompose epistemic uncertainty into two parts (inter-expert uncertainty and intra-expert uncertainty) thereby yielding interpretable uncertainty estimates with meaningful semantics.

(4) The idea of separating shared evidence from class-specific advocacy is mathematically elegant and semantically meaningful, different from the single-pass uncertainty common in EDL literature.

Weakness:

(1) While parameter overhead is quantified (0.9-1.43%), actual computation cost comparisons with baselines are missing. This is important for practical deployment.

(2) The effect of uncertainty quantification is not discussed enough. It might be clearer if the experiments include visualizations of uncertainty density on noisy data (e.g. CIFAR-10-C).

(3) The originality lies in the novel synthesis and the courtroom-inspired structure rather than entirely new mathematical objects, as MoDEX builds on existing frameworks like EDL and the EFD distribution.

---

> ### Author Rebuttal · Authors · 2026-03-30
>
> Thank you for your careful review of our paper and for the insightful and constructive comments. Please find our detailed answers to your comment below.
>
>
> **1. (Could you provide the time comparisons between MoDEX and the baseline?)**
>
> We conducted additional inference-time experiments (prediction + UQ), showing that MoDEX achieves test-time efficiency comparable to that of EDL and F-EDL while being much faster than sampling-based methods. For the detailed results and explanations, please refer to our response to **Reviewer 7Lkg’s Comment 2**.
>
> **2. (Visualizations of uncertainty)**
>
>
> We provide additional visualizations on CIFAR-10-C, showing that epistemic uncertainty increases with the corruption level ($\mathcal{C}$ = 1, 3, 5), as expected from a reliable uncertainty estimator. The results are available here: https://anonymous.4open.science/r/ICML2026_11275-D6EA/Fig1.pdf.
>
>
> **3. (Regarding the originality of MoDEX)**
>
>
> The novelty of MoDEX lies not in introducing a new mathematical primitive, but in proposing a **new structural formulation** for uncertainty-aware classification. Unlike prior EDL variants, which mostly remain alternative formulations within the EDL framework, MoDEX introduces a **courtroom-inspired decomposition and aggregation mechanism** that makes uncertainty explicit and interpretable. In this sense, MoDEX is not merely an alternative to EDL within a specific model, but a broader framework whose courtroom-based perspective may also be extended to, or provide useful interpretations for, other uncertainty-aware settings.
>
> **4. (Regarding the OOD detection metrics)**
>
> Consistent with EDL baselines, we originally reported AUPR as the primary metric and included AUROC in the Appendix.
> Following your suggestion, we additionally measured FPR95 on CIFAR-10 for OOD detection. As shown in **Table 1**, MoDEX achieves the best FPR95 among the compared methods. We will include expanded results in the appendix.
>
> **[Table 1. FPR95 results for OOD detection on CIFAR-10]**
>
> | Method|OOD. FPR95 (SVHN)↓|
> |--------|-------------|
> | EDL   | 39.90 ± 6.7 |
> | R-EDL | 32.67 ± 7.2|
> | F-EDL | 36.98 ± 7.4|
> | **MoDEX**|**28.25 ± 6.0**|
>
>
> **5. (Practical value of the decomposition of epistemic uncertainty.)**
>
> The decomposition is practically useful because the two components reflect **different sources** of uncertainty and therefore suggest **different downstream interventions** in real-world applications.
>
> High inter-expert uncertainty reflects disagreement among plausible class hypotheses, often indicating ambiguity in the decision or labeling process; this may call for refining class definitions and annotation guidelines, collecting additional annotations, or deferring the decision for human review.
>
> In contrast, high intra-expert uncertainty reflects weak or degraded evidence in the input, suggesting that the model cannot reliably interpret the given sample (e.g., due to corruption or distribution shift); this may call for improving input quality, preprocessing, or acquiring cleaner observations.
>
> **6. (Can MoDEX be generalized to non-image domains?)**
>
> **(Generalization to other domains.)** MoDEX is domain-agnostic and does not rely on image-specific structure. Following your suggestion, we additionally evaluated MoDEX on **text (AG News)** and **tabular data (UCI Adult)** under matched backbone and training protocols.  We report test accuracy and AUPR score for misclassification detection, together with expected calibration error (ECE) for AG News and AUROC for UCI Adult. **Tables 2 and 3** demonstrate that MoDEX transfers effectively beyond vision, supporting its broader applicability to uncertainty-aware classification across domains.
>
> **[Table 2. Uncertainty-aware classification on UCI Adult]**
>
> |Method|Test.Acc.↑|Miscl. AUROC ↑|Miscl. AUPR ↑|
> |------------------|-------------|----------------|---------------|
> | Softmax| 82.41 ± 0.4 | 84.88 ± 0.6| 68.38 ± 0.3   |
> | EDL | 82.40 ± 0.5 | 84.76 ± 0.5| 68.20 ± 1.6   |
> | R-EDL | 82.40 ± 0.5 | 84.77 ± 0.5| 68.23 ± 1.5|
> |F-EDL | 82.51 ± 0.4 | 85.83 ± 0.6| 70.86 ± 1.5|
> |**MoDEX**| **85.67 ± 0.4**|**91.23 ± 0.3** |**78.20 ± 0.4** |
>
>
>
> **[Table 3. Uncertainty-aware classification on AG News]**
>
> | Method|Test.Acc. ↑|Miscl. AUPR ↑|ECE ↓|
> |------------------|------------|--------|--------|
> |Softmax  |  84.80 | 80.40 | 15.15 |
> | EDL  | 88.25 | 76.15 | 16.81 |
> | R-EDL  | 86.75 | 90.76 | 12.45 |
> |F-EDL | 89.85  | 96.99 | 10.20 |
> |**MoDEX**|**90.25** |**97.69** | **5.18** |
>
> **(Potential challenges and extensions.)** A practical challenge may arise in larger-scale pretrained settings (e.g., LLM benchmarks) where fully end-to-end applications of MoDEX may be inefficient. In such cases, a natural extension is to use the pretrained model as a shared representation extractor and construct the base evidence on top of it, while fine-tuning only the lightweight class-specific advocacy and aggregation components, naturally aligning with the courtroom interpretation.

---

> > ### Author Rebuttal · Reviewer_ivYU · 2026-04-02
> >
> > The current response solves most of my problems. I will increase my score.

---

> > > ### Author Response · Authors · 2026-04-02
> > >
> > > Dear Reviewer ivYU,
> > >
> > >
> > > Thank you very much for your thoughtful re-evaluation and for revising your score. We truly appreciate your careful and constructive feedback throughout the review process. Your comments have been extremely helpful in strengthening our claims regarding computational complexity, improving our empirical evaluation (e.g., additional metrics and broader validation), and clarifying the practical implications of our uncertainty decomposition.
> > >
> > >
> > > We are very glad that our responses have addressed your concerns.
> > >
> > > Best regards,
> > >
> > > Authors

---

### Official Review · Reviewer_7Lkg · 2026-03-12

**Soundness:** 3
**Presentation:** 4
**Significance:** 3
**Originality:** 3
**Overall Recommendation:** 5
**Confidence:** 3

**Summary:**

In the paper, the idea of Evidental Deep Learning (EDL, Sensoy et al 2018) is extended by considering an ensemble of one classifier for each class label.

In EDL, a Dirichlet distribution is predicted over the class probabilities.  The proposed method aggregates the mixture of the Dirichlet distributions. The method is rigorously mathematically analyzed. Among others, EDL is proved to be a special case of the proposed method.

The main claimed advantage of the proposed method is that sampling is not needed for UQ, hence the method is computationally efficient. However, no computational resource measurements are provided.

The paper is well written. Experimentation is on well understood public data sets. The results are convincing.

**Compliance With Llm Reviewing Policy:**

Affirmed.

**Final Justification:**

The paper is well written. Experimentation is on well understood public data sets. The results are convincing. The authors answered the review questions including additional experiments. I support the acceptance of the paper.

**Key Questions For Authors:**

1 What are the computational requirements of the method? Can you justify that sampling based methods such as BNN are less efficient? How does MODEX compare computationally to EDL as the number of classes increase?

2 Can you compare the ensemble method of MODEX to that of [Balaji Lakshminarayanan, Alexander Pritzel, and Charles Blundell. Simple and scalable predictive uncertainty estimation using deep ensembles. NeurIPS 2017]? Are the methods completely unrelated? It would be nice to see an Ensemble baseline as well.

**Limitations:**

The Limitations and future directions section states that the proposed method works only for classification not regression.

Since computational efficiency is not analyzed in the paper, I wonder if large number of classes may limit efficiency.

**Strengths And Weaknesses:**

Strengths:
 - As also summarized in the Impact Statement, UQ is very important in high-stakes decision making applications and the proposed method can be an important contribution in the area.
 - Very well written paper, the main body can be understood easily. (Unfortunately I had no time to verify the proofs in the Appendix)
 - Both the theoretical analysis and the experiments are convincing.

Weaknesses:
 - Works only for classification and there seems to be no easy way of generalizing the class label based ensemble to continuous labels
 - The Introduction mentions efficiency as key feature compated to sampling based approaches. However, running time, GPU memory or any other computational requirements are never mentioned in the paper.
 - To me, the Ensemble UQ method of [Balaji Lakshminarayanan, Alexander Pritzel, and Charles Blundell. Simple and scalable predictive
uncertainty estimation using deep ensembles. NeurIPS 2017] seems relevant - altough they use ensembles very differently. The method could have been added as baseline, also in the discussion of related works.

---

> ### Author Rebuttal · Authors · 2026-03-29
>
> Thank you for your careful review of our paper and for the insightful and constructive comments. Please find our detailed answers to your comment below.
>
> **1. (Limited to classification; extension to regression is unclear.)**
>
> We agree that extending the current framework to regression is non-trivial. That said, the **broader principle behind our courtroom analogy is not inherently limited to classification** and could be extended to uncertainty-aware regression in a principled manner.
>
> One promising direction is to combine our framework with deep evidential regression (DER) (Amini et al., NeurIPS 2020) in a mixture-based formulation, where each DER expert predicts evidential regression parameters with shared and expert-specific components learned jointly. Such an extension would preserve the core spirit of our method while enabling robust UQ in regression settings, especially when targets are heterogeneous or potentially multimodal.
>
> **2. (Regarding the computational efficiency of MoDEX.)**
>
> We agree that computational efficiency is an important practical aspect, especially given our efficiency claim relative to sampling-based UQ methods. While the additional parameter overhead is outlined in Table 7 of the Appendix, inference time was not analyzed in the submission. We therefore conducted an additional inference-time comparison.
>
> **(Efficiency compared to sampling-based methods.)** We measured end-to-end inference time (prediction + UQ) against representative sampling-based UQ approaches on CIFAR-10 with VGG-16. For BNN and MC Dropout, we used 10 Monte Carlo samples, and for Deep Ensembles, 5 independently trained models. All methods were evaluated under the same setting, using RTX 4060 GPU with batch size $B=64$.
>
> **[Table 1. Computational cost comparison (CIFAR-10, VGG-16)]**
> |Method|Inference Time (ms/batch) ↓|
> |---|---:|
> |Dropout | 59.92 ± 1.5 |
> | BNN | 77.85 ± 3.5 |
> | Deep Ensembles| 34.85 ± 1.5|
> | EDL | 12.53 ± 2.4 |
> | F-EDL | 13.50 ± 0.1 |
> |MoDEX| 13.94 ± 3.1 |
>
> As shown in **Table 1**, **MoDEX is substantially more efficient than sampling-based approaches**, while remaining close to EDL and F-EDL. These support MoDEX’s practical efficiency as a single-pass UQ method.
>
> **(Scalability with respect to EDL.)** Owing to its structured decomposition, the output parameterization of MoDEX scales linearly with the number of classes, i.e., $\mathcal{O}(K)$, rather than quadratically. In practice, as the number of classes increases, the shared backbone typically dominates the total model size, so the additional capacity introduced by lightweight MoDEX heads remains relatively small and does not meaningfully hinder scalability.
>
> This is also supported by the parameter-count results: beyond Table 7, when using WideResNet-28-10 (36.5M parameters) on TinyImageNet-200, MoDEX introduces only 216K additional parameters (~0.59%). This suggests that the MoDEX remains lightweight even in higher-class settings, with relatively smaller overhead as backbone capacity increases.
> In the revised version, we will add the inference-time comparison with sampling-based methods and further clarify the scalability of MoDEX with respect to EDL.
>
>
> **3. (Can you compare MoDEX with Deep Ensembles?)**
>
> We agree that Deep Ensembles (DE) are a strong and important baseline, and we will discuss them in the *Related Work* section of the revised version.
>
> Although MoDEX is not constructed in the same manner as DE, the two are conceptually related.  In particular, Proposition 5.2 shows that MoDEX admits an ensemble-like interpretation: its predictive distribution can be written as a weighted mixture of $K$ EDL experts with class-specific evidence, while remaining a single structured model. In this sense, **MoDEX can be viewed as embedding an ensemble-style inductive bias within a single-pass architecture**.
>
> In our original experiments, we focused on single-pass UQ baselines and therefore did not include DE. Nevertheless, we agree that a direct comparison with DE is valuable and will include it in the appendix of the revised version. For completeness, we report the preliminary CIFAR-10 results in **Table 2**. While DE achieves slightly higher ID accuracy and misclassification detection performance, **MoDEX outperforms DE on OOD detection with only a single forward pass**, showing a favorable efficiency-UQ trade-off.
>
> **[Table 2. Comparison with DE using 5 independently trained models on CIFAR-10]**
> | Method| Test Acc. ↑| Miscl. AUPR. ↑| SVHN ↑| C-100 ↑|
> |---------------------------|--------------------|--------------------|--------------------|-------------------|
> | DE (M=5)|**92.55 ± 0.1**| **99.32 ± 0.0**| 84.77 ± 1.4| 89.15 ± 0.2|
> |MoDEX|92.46 ± 0.2| 99.18 ± 0.0| **91.58 ± 0.4** | **89.28 ± 0.3**|
>
>
> **4. (Regarding the source code.)**
>
> We have already included the source code in the *supplementary material* submitted with the paper. We would be grateful if the reviewer could kindly check the supplementary files.

---

> > ### Author Rebuttal · Reviewer_7Lkg · 2026-04-02
> >
> > Aplogies, the Openreview UI is so much overloaded that I just did not see the supplementary materials link.
> >
> > My questions are answered. As I was already positive about the paper, I maintain my "Accept" recommendation.

---

> > > ### Author Response · Authors · 2026-04-04
> > >
> > > Dear Reviewer 7Lkg,
> > >
> > > Thank you very much for maintaining your positive rating. We sincerely appreciate your thoughtful and constructive feedback throughout the review process.
> > >
> > > Your comments have been particularly valuable in helping us further strengthen the paper. In the revised version, we will incorporate a detailed computational efficiency analysis, provide a clearer comparison with sampling-based methods such as BNN, and include explicit comparisons with Deep Ensembles. We will also further discuss the potential extension of our framework to regression settings.
> > >
> > > We are very glad that our responses have addressed your concerns.
> > >
> > > Best regards,
> > >
> > > Authors

---

### Decision · Program_Chairs · 2026-04-30

**Decision:**

Accept (regular)

**Comment:**

Reviewers agreed that this paper is technically sound, clearly written, and presents an interesting and interpretable extension of evidential deep learning. In particular, they appreciated the structured MoDEX formulation, the courtroom-inspired decomposition of shared and class-specific evidence, and the strong empirical performance across uncertainty-aware classification benchmarks.

The rebuttal addressed the main initial concerns. In particular, it clarified the computational profile relative to sampling based methods, added comparisons to Deep Ensembles and recent non-EDL single-pass baselines, expanded the empirical picture with additional metrics and non-vision results, and better positioned the method with respect to known limitations of EDL. These clarifications appear to have resolved the main reviewer concerns, including for the reviewer who initially raised the strongest conceptual objection.

Overall, while some limitations remain, most notably that the current framework is restricted to classification and that some of the interpretability semantics are best viewed as an inductive bias rather than literal supervision, reviewers no longer viewed these as blocking issues. I therefore recommend acceptance, as the paper makes a meaningful and well-supported contribution to single-pass uncertainty-aware classification.